# Learning to Correlate in Multi-Player General-Sum Sequential Games

**Andrea Celli**[*]
Politecnico di Milano
andrea.celli@polimi.it

**Alberto Marchesi**[*]
Politecnico di Milano
alberto.marchesi@polimi.it

**Tommaso Bianchi**
Politecnico di Milano
tommaso4.bianchi@mail.polimi.it

**Nicola Gatti**
Politecnico di Milano
nicola.gatti@polimi.it

## Abstract

In the context of multi-player, general-sum games, there is a growing interest in solution concepts involving some form of communication among players, since they can lead to socially better outcomes with respect to Nash equilibria and may be reached through learning dynamics in a decentralized fashion. In this paper, we focus on *coarse correlated equilibria* (CCEs) in sequential games. First, we complete the picture on the complexity of finding social-welfare-maximizing CCEs by proving that the problem is not in Poly-$\mathsf{APX}$, unless $\mathsf{P} = \mathsf{NP}$, in games with three or more players (including chance). Then, we provide simple arguments showing that CFR—working with behavioral strategies—may not converge to a CCE in multi-player, general-sum sequential games. In order to amend this issue, we devise two variants of CFR that provably converge to a CCE. The first one (CFR-S) is a simple stochastic adaptation of CFR which employs sampling to build a correlated strategy, whereas the second variant (called CFR-Jr) enhances CFR with a more involved reconstruction procedure to recover correlated strategies from behavioral ones. Experiments on a rich testbed of multi-player, general-sum sequential games show that both CFR-S and CFR-Jr are dramatically faster than the state-of-the-art algorithms to compute CCEs, with CFR-Jr being also a good heuristic to find socially-optimal CCEs.

## 1 Introduction

A number of recent studies explore relaxations of the classical notion of equilibrium (*i.e.*, the *Nash equilibrium* (NE) [30]), allowing to model communication among the players [3, 14, 34]. Communication naturally brings about the possibility of playing correlated strategies. These are customarily modeled through a trusted external mediator who privately recommends actions to the players [1]. In particular, a correlated strategy is a *correlated equilibrium* (CE) if each player has no incentive to deviate from the recommendation, assuming the other players would not deviate either. A popular variation of the CE is the *coarse correlated equilibrium* (CCE), which only prevents deviations before knowing the recommendation [28]. In sequential games, CEs and CCEs are well-suited for scenarios where the players have limited communication capabilities and can only communicate before the game starts, such as, *e.g.*, military settings where field units have no time or means of communicating during a battle, collusion in auctions where communication is illegal during bidding, and, in general, any setting with costly communication channels or blocking environments.

---

[*]Equal contribution.

CCEs present a number of appealing properties. A CCE can be reached through simple (no-regret) learning dynamics in a decentralized fashion [17, 22], and, in several classes of games (such as, *e.g.*, normal-form and succinct games [31, 26]), it can be computed exactly in time polynomial in the size of the input. Furthermore, an optimal (*i.e.*, social-welfare-maximizing) CCE may provide arbitrarily larger welfare than an optimal CE, which, in turn, may provide arbitrarily better welfare than an optimal NE [12]. Although the problem of finding an optimal CCE is NP-hard for some game classes (such as, *e.g.*, graphical, polymatrix, congestion, and anonymous games [3]), Roughgarden [34] shows that the CCEs reached through regret-minimizing procedures have near-optimal social welfare when the $(\lambda, \mu)$-smoothness condition holds. This happens, *e.g.*, in some specific auctions, congestion games, and even in Bayesian settings, as showed by Hartline et al. [23]. Thus, decentralized computation via learning dynamics, computational efficiency, and welfare optimality make the CCE one of the most interesting solution concepts for practical applications. However, the problem of computing CCEs has been addressed only for some specific games with particular structures [3, 23]. In this work, we study how to compute CCEs in the general class of games which are sequential, general-sum, and multi-player. This is a crucial advancement of CCE computation, as sequential games provide a model for strategic interactions which is richer and more adherent to real-world situations than the normal form.

In sequential games, it is known that, when there are two players without chance moves, an optimal CCE can be computed in polynomial time [12]. Celli et al. [12] also provide an algorithm (with no polynomiality guarantees) to compute solutions in multi-player games, using a column-generation procedure with a MILP pricing oracle. As for computing approximate CCEs, in the normal-form setting, any *Hannan consistent* regret-minimizing procedure for simplex decision spaces may be employed to approach the set of CCEs [5, 13]—the most common of such techniques is *regret matching* (RM) [4, 22]. However, approaching the set of CCEs in sequential games is more demanding. One could represent the sequential game with its equivalent normal form and apply RM to it. However, this would result in a guarantee on the cumulative regret which would be exponential in the size of the game tree (see Section 2). Thus, reaching a good approximation of a CCE could require an exponential number of iterations. The problem of designing learning algorithms avoiding the construction of the normal form has been successfully addressed in sequential games for the two-player, zero-sum setting. This is done by decomposing the overall regret locally at the information sets of the game [15]. The most widely adopted of such approaches are *counterfactual regret minimization* (CFR) [43] and CFR+ [39, 38], which originated variants such as those introduced by Brown and Sandholm [9] and Brown et al. [10]. These techniques were the key for many recent remarkable results [6, 7, 8**?** ]. However, these algorithms work with players' behavioral strategies rather than with correlated strategies, and, thus, they are not guaranteed to approach CCEs in general-sum games, even with two players. The only known theoretical guarantee of CFR when applied to multi-player, general-sum games is that it excludes dominated actions [19]. Some works also attempt to apply CFR to multi-player, zero-sum games, see, *e.g.*, [32].

**Original contributions**  First, we complete the picture on the computational complexity of finding an optimal CCE in sequential games, showing that the problem is inapproximable (*i.e.*, not in Poly-APX), unless P = NP, in games with three or more players (chance included). In the rest of the paper, we focus on how to compute approximate CCEs in multi-player, general-sum, sequential games using no-regret-learning procedures. We start pointing out simple examples where CFR-like algorithms available in the literature cannot be directly employed to our purpose, as they only provide players' average behavioral strategies whose product is not guaranteed to converge to an approximate CCE. However, we show how CFR can be easily adapted to approach the set of CCEs in multi-player, general-sum sequential games by resorting to sampling procedures (we call the resulting, naïve algorithm CFR-S). Then, we design an enhanced version of CFR (called CFR-Jr) which computes an average correlated strategy guaranteed to converge to an approximate CCE with a bound on the regret sub-linear in the size of the game tree. The key component of CFR-Jr is a polynomial-time algorithm which constructs, at each iteration, the players' normal-form strategies by working on the game tree, avoiding to build the (exponential-sized) normal-form representation. We evaluate the scalability of CFR-S and CFR-Jr on a rich testbed of multi-player, general-sum sequential games. Both algorithms solve instances which are orders of magnitude larger than those solved by previous state-of-the-art CCE-finding techniques. Moreover, CFR-Jr proved to be a good heuristic to compute optimal CCEs, returning nearly-socially-optimal solutions in all the instances of our testbeds. Finally, we also test our algorithms against CFR in multi-player, general-sum games, showing that, in several

instances of our testbed, CFR does not converge to a CCE and it returns solutions providing a social welfare considerably lower than that achieved with CFR-S and CFR-Jr.

## 2 Preliminaries

In this section, we introduce some basic concepts which are used in the rest of the paper (see Shoham and Leyton-Brown [36] and Cesa-Bianchi and Lugosi [13] for further details).

### 2.1 Extensive-form games and relevant solution concepts

We focus on *extensive-form games* (EFGs) with imperfect information and perfect recall. We denote the set of players as $\mathcal{P} \cup \{c\}$, where $c$ is the *Nature* (*chance*) player (representing exogenous stochasticity) selecting actions with a fixed known probability distribution. $H$ is the set of nodes of the game tree, and a node $h \in H$ is identified by the ordered sequence of actions from the root to the node. $Z \subseteq H$ is the set of terminal nodes, which are the leaves of the game tree. For every $h \in H \setminus Z$, we let $P(h)$ be the unique player who acts at $h$ and $A(h)$ be the set of actions she has available. We write $h \cdot a$ to denote the node reached when $a \in A(h)$ is played at $h$. For each player $i \in \mathcal{P}$, $u_i : Z \to \mathbb{R}$ is the payoff function. We denote by $\Delta$ the maximum range of payoffs in the game, *i.e.*, $\Delta = \max_{i \in \mathcal{P}} (\max_{z \in Z} u_i(z) - \min_{z \in Z} u_i(z))$.

We represent imperfect information using *information sets* (from here on, infosets). Any infoset $I$ belongs to a unique player $i$, and it groups nodes which are indistinguishable for that player, *i.e.*, $A(h) = A(h')$ for any pair of nodes $h, h' \in I$. $\mathcal{I}_i$ denotes the set of all player $i$'s infosets, which form a partition of $\{h \in H \mid P(h) = i\}$. We denote by $A(I)$ the set of actions available at infoset $I$. In perfect-recall games, the infosets are such that no player forgets information once acquired.

We denote with $\pi_i$ a *behavioral strategy* of player $i$, which is a vector defining a probability distribution at each player $i$'s infoset. Given $\pi_i$, we let $\pi_{i,I}$ be the (sub)vector representing the probability distribution at $I \in \mathcal{I}_i$, with $\pi_{i,I,a}$ denoting the probability of choosing action $a \in A(I)$.

An EFG has an equivalent tabular (*normal-form*) representation. A *normal-form plan* for player $i$ is a vector $\sigma_i \in \Sigma_i = \bigtimes_{I \in \mathcal{I}_i} A(I)$ which specifies an action for each player $i$'s infoset. Then, an EFG is described through a $|\mathcal{P}|$-dimensional matrix specifying a utility for each player at each *joint normal-form plan* $\sigma \in \Sigma = \bigtimes_{i \in \mathcal{P}} \Sigma_i$. The expected payoff of player $i$, when she plays $\sigma_i \in \Sigma_i$ and the opponents play normal-form plans in $\sigma_{-i} \in \Sigma_{-i} = \bigtimes_{j \neq i \in \mathcal{P}} \Sigma_j$, is denoted, with an overload of notation, by $u_i(\sigma_i, \sigma_{-i})$. Finally, a *normal-form strategy* $x_i$ is a probability distribution over $\Sigma_i$. We denote by $\mathcal{X}_i$ the set of the normal-form strategies of player $i$. Moreover, $\mathcal{X}$ denotes the set of joint probability distributions defined over $\Sigma$.

We also introduce the following notation. We let $\rho^{\pi_i}$ be a vector in which each component $\rho_z^{\pi_i}$ is the probability of reaching the terminal node $z \in Z$, given that player $i$ adopts the behavioral strategy $\pi_i$ and the other players play so as to reach $z$. Similarly, given a normal-form plan $\sigma_i \in \Sigma_i$, we define the vector $\rho^{\sigma_i}$. Moreover, with an abuse of notation, $\rho_I^{\pi_i}$ and $\rho_I^{\sigma_i}$ denote the probability of reaching infoset $I \in \mathcal{I}_i$. Finally, $Z(\sigma_i) \subseteq Z$ is the subset of terminal nodes which are (potentially) reachable if player $i$ plays according to $\sigma_i \in \Sigma_i$.

The classical notion of CE by Aumann [1] models correlation via the introduction of an external mediator who, before the play, draws the joint normal-form plan $\sigma^* \in \Sigma$ according to a publicly known $x^* \in \mathcal{X}$, and privately communicates each *recommendation* $\sigma_i^*$ to the corresponding player. After observing their recommended plan, each player decides whether to follow it or not. A CCE is a relaxation of the CE, defined by Moulin and Vial [28], which enforces protection against deviations which are independent from the sampled joint normal-form plan.

**Definition 1.** *A CCE of an EFG is a probability distribution $x^* \in \mathcal{X}$ such that, for every $i \in \mathcal{P}$, and $\sigma_i' \in \Sigma_i$, it holds:*

$$\sum_{\sigma_i \in \Sigma_i} \sum_{\sigma_{-i} \in \Sigma_{-i}} x^*(\sigma_i, \sigma_{-i}) \left( u_i(\sigma_i, \sigma_{-i}) - u_i(\sigma_i', \sigma_{-i}) \right) \geq 0.$$

CCEs differ from CEs in that a CCE only requires that following the suggested plan is a best response in expectation, before the recommended plan is actually revealed. In both equilibrium concepts, the

entire probability distribution according to which recommendations are drawn is revealed before the game starts. After that, each player commits to playing a normal-form plan (see Appendix A for further details on the various notions of correlated equilibrium in EFGs). An NE [30] is a CCE which can be written as a product of players' normal-form strategies $x_i^* \in \mathcal{X}_i$. In conclusion, an $\varepsilon$-CCE is a relaxation of a CCE in which every player has an incentive to deviate less than or equal to $\varepsilon$ (the same definition holds true for $\varepsilon$-CE and $\varepsilon$-NE).

## 2.2  Regret and regret minimization

In the *online convex optimization* framework [42], each player $i$ plays repeatedly against an unknown environment by making a series of decisions $x_i^1, x_i^2, \ldots, x_i^t$. In the basic setting, the decision space of player $i$ is the whole normal-form strategy space $\mathcal{X}_i$. At iteration $t$, after selecting $x_i^t$, player $i$ observes a utility $u_i^t(x_i^t)$. The *cumulative external regret* of player $i$ up to iteration $T$ is defined as

$$R_i^T = \max_{\hat{x}_i \in \mathcal{X}_i} \sum_{t=1}^{T} u_i^t(\hat{x}_i) - \sum_{t=1}^{T} u_i^t(x_i^t). \tag{1}$$

A *regret minimizer* is a function providing the next player $i$'s strategy $x_i^{t+1}$ on the basis of the past history of play and the observed utilities up to iteration $t$. A desirable property for regret minimizers is *Hannan consistency* [21], which requires that $\limsup_{T \to \infty} \frac{1}{T} R_i^T \leq 0$, *i.e.*, the cumulative regret grows at a sublinear rate in the number of iterations $T$.

In an EFG, the regret can be defined at each infoset. After $T$ iterations, the cumulative regret for not having selected action $a \in A(I)$ at infoset $I \in \mathcal{I}_i$ (denoted by $R_I^T(a)$) is the cumulative difference in utility that player $i$ would have experienced by selecting $a$ at $I$ instead of following the behavioral strategy $\pi_i^t$ at each iteration $t$ up to $T$. Then, the regret for player $i$ at infoset $I \in \mathcal{I}_i$ is defined as $R_I^T = \max_{a \in A(I)} R_I^T(a)$. Moreover, we let $R_I^{T,+}(a) = \max\{R_I^T(a), 0\}$.

RM [22] is the most widely adopted regret-minimizing scheme when the decision space is $\mathcal{X}_i$ (*e.g.*, in normal-form games). In the context of EFGs, RM is usually applied locally at each infoset, where the player selects a distribution over available actions proportionally to their positive regret. Specifically, at iteration $T+1$ player $i$ selects actions $a \in A(I)$ according to the following probability distribution:

$$\pi_{i,I,a}^{T+1} = \begin{cases} \frac{R_I^{T,+}(a)}{\sum_{a' \in A(I)} R_I^{T,+}(a')}, & \text{if } \sum_{a' \in A(I)} R_I^{T,+}(a') > 0 \\ \frac{1}{|A(I)|}, & \text{otherwise} \end{cases}.$$

Playing according to RM at each iteration guarantees, on iteration $T$, $R_I^T \leq \Delta \frac{\sqrt{|A(I)|}}{\sqrt{T}}$ [13]. CFR [43] is an anytime algorithm to compute $\varepsilon$-NEs in two-player, zero-sum EFGs. CFR minimizes the external regret $R_i^T$ by employing RM locally at each infoset. In two-player, zero-sum games, if both players have cumulative regrets such that $\frac{1}{T} R_i^T \leq \varepsilon$, then their average behavioral strategies are a $2\varepsilon$-NE [41]. CFR+ is a variation of classical CFR which exhibits better practical performances [39]. However, it uses alternation (*i.e.*, it alternates which player updates her regret on each iteration), which complicates the theoretical analysis to prove convergence [15, 11].

## 3  Hardness of approximating optimal CCEs

We address the following question: given an EFG, can we find a social-welfare-maximizing CCE in polynomial time? As shown by Celli et al. [12], the answer is yes in two-player EFGs without Nature. Here, we give a negative answer to the question in the remaining cases, *i.e.*, two-player EFGs with Nature (Theorem 1) and EFGs with three or more players without Nature (Theorem 2). Specifically, we provide an even stronger negative result: there is no polynomial-time approximation algorithm which finds a CCE whose value approximates that of a social-welfare-maximizing CCE up to any polynomial factor in the input size, unless $\mathsf{P} = \mathsf{NP}$. [2] We prove our results by means of a reduction from SAT, a well known $\mathsf{NP}$-complete problem [18], which reads as follows.

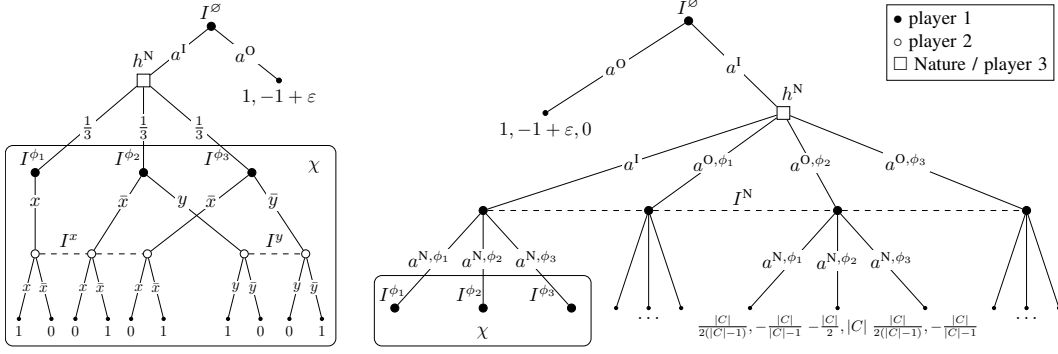

Figure 1: *Left*: Example of game for the reduction of Theorem 1, where $V = \{x, y, z\}$, $C = \{\phi_1, \phi_2, \phi_3\}$, $\phi_1 = x$, $\phi_2 = \bar{x} \vee y$, and $\phi_3 = \bar{x} \vee \bar{y}$. *Right*: Example of game for the reduction of Theorem 2, with $V$ and $C$ as before.

**Definition 2** (SAT). *Given a finite set $C$ of clauses defined over a finite set $V$ of variables, is there a truth assignment to the variables which satisfies all clauses?*

For clarity, Figure 1 shows concrete examples of the EFGs employed for the reductions of Theorems 1 and 2. Here, we only provide proof sketches, while we report full proofs in Appendix B.

**Theorem 1.** *Given a two-player EFG with Nature, the problem of computing a social-welfare-maximizing CCE is not in Poly-*APX *unless* P = NP. [3]

*Proof sketch.* An example of our reduction from SAT is provided on the left of Figure 1. Its main idea is the following: player 2 selects a truth assignment to the variables, while player 1 chooses a literal for each clause in order to satisfy it. It can be proved that there exists a CCE in which each player gets utility 1 if and only if SAT is satisfiable (as player 1 selects $a^{\mathrm{I}}$), otherwise player 1 plays $a^{\mathrm{O}}$ in any CCE and its social welfare is $\varepsilon$. Assume there is a a polynomial-time $\mathsf{poly}(\eta)$-approximation algorithm $\mathcal{A}$. If SAT is satisfiable, $\mathcal{A}$ would return a CCE with social welfare at least $\frac{2}{\mathsf{poly}(\eta)}$. Since, for $\eta$ sufficiently large it holds $\frac{2}{\mathsf{poly}(\eta)} > \frac{1}{2^\eta}$, then $\mathcal{A}$ would allow us to decide in polynomial time whether SAT is satisfiable, leading to a contradiction unless P = NP. □

**Theorem 2.** *Given a three-player EFG without Nature, the problem of computing a social-welfare-maximizing CCE is not in Poly-*APX *unless* P = NP.

*Proof sketch.* An example of our reduction from SAT is provided on the right of Figure 1. It is based on the same idea as that of the previous proof, where the uniform probability distribution played by Nature is simulated by a particular game gadget (requiring a third player). □

## 4 CFR in multi-player general-sum sequential games

In this section, we first highlight why CFR cannot be directly employed when computing CCEs of general-sum games. Then, we show a simple way to amend it.

### 4.1 Convergence to CCEs in general-sum games

When players follow strategies recommended by a regret minimizer, the *empirical frequency of play* approaches the set of CCEs [13]. Suppose that, at time $t$, the players play a joint normal-form plan $\sigma^t \in \Sigma$ drawn according to their current strategies. Then, the empirical frequency of play after $T$ iterations is defined as the joint probability distribution $\bar{x}^T \in \mathcal{X}$ such that $\bar{x}^T(\sigma) \coloneqq \frac{|t \leq T : \sigma^t = \sigma|}{T}$ for every $\sigma \in \Sigma$. However, vanilla CFR and its most popular variations (such as, *e.g.*, CFR+ [39] and DCFR [9]) do not keep track of the empirical frequency of play, as they only keep track of the players'

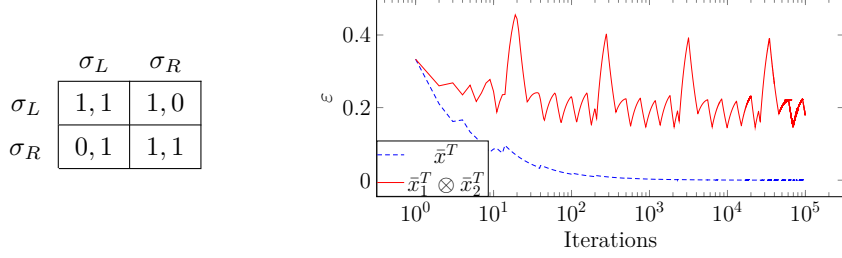

| | $\sigma_L$ | $\sigma_R$ |
|---|---|---|
| $\sigma_L$ | $1,1$ | $1,0$ |
| $\sigma_R$ | $0,1$ | $1,1$ |

Figure 2: *Left*: Game where $\bar{x}_1^T \otimes \bar{x}_2^T$ does not converge to a CCE. *Right*: Approximation attained by $\bar{x}^T$ and $\bar{x}_1^T \otimes \bar{x}_2^T$ when RM is applied to a variation of the Shapley game (see Appendix C).

average behavioral strategies. This ensures that the strategies are compactly represented, but it is not sufficient to recover a CCE in multi-player, general-sum games. Indeed, it is possible to show that, even in normal-form games, if the players play according to some regret-minimizing strategies, then the product distribution $x \in \mathcal{X}$ resulting from players' (marginal) average strategies may not converge to a CCE. In order to see this, we provide the following simple example.

**Example** Consider the two-player normal-form game depicted on the left in Figure 2. At iteration $t$, let players' strategies $x_1^t, x_2^t$ be such that $x_1^t(\sigma_L) = x_2^t(\sigma_L) = (t+1) \mod 2$. Clearly, $u_1^t(x_1^t) = u_2^t(x_2^t) = 1$ for any $t$. For both players, at iteration $t$, the regret of not having played $\sigma_L$ is 0, while the regret of $\sigma_R$ is $-1$ if and only if $t$ is even, otherwise it is 0. As a result, after $T$ iterations, $R_1^T = R_2^T = 0$, and, thus, $x_1^t$ and $x_2^t$ minimize the cumulative external regret. Players' average strategies $\bar{x}_1^T = \frac{1}{T} \sum_{t=1}^T x_1^t$ and $\bar{x}_2^T = \frac{1}{T} \sum_{t=1}^T x_2^t$ converge to $(\frac{1}{2}, \frac{1}{2})$ as $T \to \infty$. However, $x \in \mathcal{X}$ such that $x(\sigma) = \frac{1}{4}$ for every $\sigma \in \Sigma$ is not a CCE of the game. Indeed, a player is always better off playing $\sigma_L$, obtaining a utility of 1, while she only gets $\frac{3}{4}$ if she chooses to stick to $x$. We remark that $\bar{x}^T$ converges, as $T \to \infty$, to $x \in \mathcal{X} : x(\sigma_L, \sigma_L) = x(\sigma_R, \sigma_R) = \frac{1}{2}$, which is a CCE.

The example above employs handpicked regret-minimizing strategies, but similar examples can be easily found when applying common regret minimizers. As an illustrative case, Figure 2 shows, on the right, that, even with a simple variation of the Shapley game (see Appendix C), the outer product of the average strategies $\bar{x}_1^T \otimes \bar{x}_2^T$ obtained via RM does not converge to a CCE as $T \to \infty$. It is clear that the same issue may (and does, see Figures 3 and 5) happen when directly applying CFR to general-sum EFGs.

## 4.2 CFR with sampling (CFR-S)

---
**Algorithm 1** CFR-S for player $i$

---
1: **function** CFR-S($\Gamma$,$i$)
2:     Initialize regret minimizer for each $I \in \mathcal{I}_i$
3:     $t \leftarrow 1$
4:     **while** $t < T$ **do**
5:         $\sigma_i^t \leftarrow$ RECOMMEND($I_\varnothing$)
6:         Observe $u_i^t(\sigma_i) \coloneqq u_i(\sigma_i, \sigma_{-i}^t)$
7:         UPDATE($I_\varnothing, \sigma_i^t, u_i^t$)
8:         $t \leftarrow t + 1$

---

Motivated by the previous examples, we describe a simple variation of CFR guaranteeing approachability to the set of CCEs even in multi-player, general-sum EFGs. Vanilla CFR proceeds as follows (see Subsection 2.2 for the details): for each iteration $t$, and for each infoset $I \in \mathcal{I}_i$, player $i$ observes the realized utility for each action $a \in A(I)$, and then computes $\pi_{i,I}^t$ according to standard RM. Once $\pi_{i,I}^t$ has been computed, it is used by the regret minimizers of infosets on the path from the root to $I$ so as to compute observed utilities. We propose CFR *with sampling* (CFR-S) as a simple way to keep track of the empirical frequency of play. The basic idea is letting each player $i$, at each $t$, draw $\sigma_i^t$ according to her current strategy. Algorithm 1 describes the structure of CFR-S, where function RECOMMEND builds a normal-form plan $\sigma_i^t$ by sampling, at each $I \in \mathcal{I}_i$, an action in $A(I)$ according to $\pi_i^t$ computed via RM, and UPDATE updates the average regrets local to each regret minimizer by propagating utilities according to $\sigma_i^t$. Each player $i$ experiences utilities depending, at each $t$, on the sampled plans $\sigma_{-i}^t$ (Line 6). Joint normal form plans $\sigma^t \coloneqq (\sigma_i^t, \sigma_{-i}^t)$ can be easily stored to compute the empirical frequency of play. We state the following (see Appendix C for detailed proofs):

**Theorem 3.** *The empirical frequency of play $\bar{x}^T$ obtained with CFR-S converges to a CCE almost surely, for $T \to \infty$.*

Moreover, the cumulative regret grows as $O(T^{-1/2})$. This result is in line with the approach of Hart and Mas-Colell [22] in normal-form games. Despite its simplicity, we show (see Section 6 for an experimental evaluation) that it is possible to achieve better perfomances via a smarter reconstruction technique that keeps CFR deterministic, avoiding any sampling step.

## 5 CFR with joint distribution reconstruction (CFR-Jr)

We design a new method—called *CFR with joint distribution reconstruction* (CFR-Jr)—to enhance CFR so as to approach the set of CCEs in multi-player, general-sum EFGs. Differently from the naïve CFR-S algorithm, CFR-Jr does not sample normal-form plans, thus avoiding any stochasticity.

The main idea behind CFR-Jr is to keep track of the average joint probability distribution $\bar{x}^T \in \mathcal{X}$ arising from the regret-minimizing strategies built with CFR. Formally, $\bar{x}^T = \frac{1}{T} \sum_{t=1}^{T} x^t$, where $x^t \in \mathcal{X}$ is the joint probability distribution defined as the product of the players' normal-form strategies at iteration $t$. At each $t$, CFR-Jr computes $\pi_i^t$ with CFR's update rules, and then constructs a strategy $x_i^t \in \mathcal{X}_i$ which is realization equivalent (*i.e.*, it induces the same probability distribution on the terminal nodes, see [36] for a formal definition) to $\pi_i^t$. We do this efficiently by directly working on the game tree, without resorting to the normal-form representation. Strategies $x_i^t$ are then employed to compute $x^t$. The pseudocode of CFR-Jr is provided in Appendix D.

---

**Algorithm 2** Reconstruct $x_i$ from $\pi_i$

---

1: **function** NF-STRATEGY-RECONSTRUCTION($\pi_i$)
2:   $\mathbf{X} \leftarrow \varnothing$   ▷ $\mathbf{X}$ is a dictionary defining $x_i$
3:   $\omega_z \leftarrow \rho_z^{\pi_i} \quad \forall z \in Z$
4:   **while** $\omega > 0$ **do**
5:     $\bar{\sigma}_i \leftarrow \arg\max_{\sigma_i \in \Sigma_i} \min_{z \in Z(\sigma_i)} \omega_z$
6:     $\bar{\omega} \leftarrow \min_{z \in Z(\bar{\sigma}_i)} \omega_z$
7:     $\mathbf{X} \leftarrow \mathbf{X} \cup (\bar{\sigma}_i, \bar{\omega})$
8:     $\omega \leftarrow \omega - \bar{\omega} \rho^{\bar{\sigma}_i}$
    **return** $x_i$ built from the pairs in $\mathbf{X}$

---

Algorithm 2 shows a polynomial-time procedure to compute a normal-form strategy $x_i \in \mathcal{X}_i$ realization equivalent to a given behavioral strategy $\pi_i$. The algorithm maintains a vector $\omega$ which is initialized with the probabilities of reaching the terminal nodes by playing $\pi_i$ (Line 3), and it works by iteratively assigning probability to normal-form plans so as to induce the same distribution as $\omega$ over $Z$. [4] In order for this to work, at each iteration, the algorithm must pick a normal-form plan $\bar{\sigma}_i \in \Sigma_i$ which maximizes the minimum (remaining) probability $\omega_z$ over the terminal nodes $z \in Z(\bar{\sigma}_i)$ reachable when playing $\bar{\sigma}_i$ (Line 5). Then, the probabilities $\omega_z$ for $z \in Z(\bar{\sigma}_i)$ are decreased by the minimum (remaining) probability $\bar{\omega}$ corresponding to $\bar{\sigma}_i$, and $\bar{\sigma}_i$ is assigned probability $\bar{\omega}$ in $x_i$. The algorithm terminates when the vector $\omega$ is zeroed, returning a normal-form strategy $x_i$ realization equivalent to $\pi_i$. This is formally stated by the following result, which also provides a polynomial (in the size of the game tree) upper bound on the run time of the algorithm and on the support size of the returned normal-form strategy $x_i$. [5]

**Theorem 4.** *Algorithm 2 outputs a normal-form strategy $x_i \in \mathcal{X}_i$ realization equivalent to a given behavioral strategy $\pi_i$, and it runs in time $O(|Z|^2)$. Moreover, $x_i$ has support size at most $|Z|$.*

Intuitively, the result in Theorem 4 (its full proof is in Appendix D) relies on the crucial observation that, at each iteration, there is at least one terminal node $z \in Z$ whose corresponding probability $\omega_z$ is zeroed during that iteration. The algorithm is guaranteed to terminate since each $\omega_z$ is never negative, which is the case given how the normal-form plans are selected (Line 5), and since the game has perfect recall. This guarantees that the algorithm eventually terminates in at most $|Z|$ iterations.

Finally, the following theorem (whose full proof is in Appendix D) proves that the average distribution $\bar{x}^T$ obtained with CFR-Jr approaches the set of CCEs. Formally:

**Theorem 5.** *If $\frac{1}{T} R_i^T \leq \varepsilon$ for each player $i \in \mathcal{P}$, then $\bar{x}^T$ obtained with CFR-Jr is an $\varepsilon$-CCE.*

This is a direct consequence of the connection between regret-minimizing procedures and CCEs, and of the fact that $\bar{x}^T$ is obtained by averaging the products of normal-form strategies which are equivalent to regret-minimizing behavioral strategies obtained with CFR.

Table 1: Comparison between the run time and the social welfare of CFR-S, CFR-Jr (for various levels of accuracy $\alpha$), and the CG algorithm. General-sum instances are marked with $\star$. Results of CFR-S are averaged over 50 runs. We generated 20 instances for each R$p$-$d$ family. $> 24$h means that the execution of the algorithm was stopped before its completion, after running for 24 hours.

| Game | Tree size #infosets | CFR-S | | | | CFR-Jr | | | | CG |
|---|---|---|---|---|---|---|---|---|---|---|
| | | $\alpha = 0.05$ | $\alpha = 0.005$ | $\alpha = 0.0005$ | $\text{sw}_{\text{APX}}/\text{sw}_{\text{OPT}}$ | $\alpha = 0.05$ | $\alpha = 0.005$ | $\alpha = 0.0005$ | $\text{sw}_{\text{APX}}/\text{sw}_{\text{OPT}}$ | |
| K3-6 | 72 | 1.41s | 9h15m | $> 24$h | - | 1.03s | 13.41s | 11m21s | - | 3h47m |
| K3-7 | 84 | 4.22s | 17h11m | $> 24$h | - | 2.35s | 14.33s | 51m27s | - | 14h37m |
| K3-10 | 120 | 22.69s | $> 24$h | $> 24$h | - | 7.21s | 72.78s | 4h11m | - | $> 24$h |
| L3-4 | 1200 | 10m33s | $> 24$h | $> 24$h | - | 1m15s | 6h10s | $> 24$h | - | $> 24$h |
| L3-6 | 2664 | 2h5m | $> 24$h | $> 24$h | - | 2m40s | 11h19m | $> 24$h | - | $> 24$h |
| L3-8 | 4704 | 13h55m | $> 24$h | $> 24$h | - | 20m22s | $> 24$h | $> 24$h | - | $> 24$h |
| G2-4-A$^\star$ | 4856 | 10m31s | $> 24$h | $> 24$h | 0.979 | 20m23m | 11h4m | $> 24$h | 0.994 | $> 24$h |
| G2-4-DA$^\star$ | 4856 | 2m1s | 3h28m | 4h17m | 0.918 | 1m36 | 56m6s | $> 24$h | 0.976 | $> 24$h |
| G2-4-DH$^\star$ | 4856 | 1m19s | 2h7m | 3h28m | 0.918 | 1m51s | 1h5m | $> 24$h | 0.976 | $> 24$h |
| G2-4-AL$^\star$ | 4856 | 2m3s | 1h33m | 4h20m | 0.919 | 1m48s | 55m43s | $> 24$h | 0.976 | $> 24$h |
| G3-4-A$^\star$ | 98508 | 1h33m | $> 24$h | $> 24$h | 0.996 | 1h3m | 4h13m | $> 24$h | 0.999 | $> 24$h |
| G3-4-DA$^\star$ | 98508 | 1h13m | $> 24$h | $> 24$h | 0.987 | 12m18s | 1h50m | $> 24$h | 1.000 | $> 24$h |
| G3-4-DH$^\star$ | 98508 | 47m33s | 19h40m | $> 24$h | 0.886 | 16m38s | 4h8m | 15h27m | 1.000 | $> 24$h |
| G3-4-AL$^\star$ | 98508 | 32m34s | 15h32m | 17h30m | 0.692 | 1h21m | 5h2s | $> 24$h | 0.730 | $> 24$h |
| R3-12$^\star$ | 3071 | 1m44s | 35m38s | 3h8m | 0.907 | 16.94s | 3m19s | 24m6s | 0.897 | $> 24$h |
| R3-15$^\star$ | 24542 | 21m30s | 4h28m | 7h50m | 0.924 | 3m34s | 14m53s | 3h3m | 0.931 | $> 24$h |

## 6 Experimental evaluation

We experimentally evaluate CFR-Jr, comparing its performance with that of CFR-S, CFR, and the state-of-the-art algorithm for computing optimal CCEs (denoted by CG) [12]. [6] This algorithm is a variation of the simplex method employing a column generation technique based on a MILP pricing oracle (we use the GUROBI 8.0 MILP solver). Notice that directly applying RM on the normal form is not feasible, as $|\Sigma| > 10^{20}$ even for the smallest instances. Further results are in Appendix E.

**Setup** We conduct experiments on parametric instances of three-player Kuhn poker games [27], three-player Leduc hold'em poker games [37], two/three-player Goofspiel games [33], and some randomly generated general-sum EFGs. The two-player, zero-sum versions of these games are standard benchmarks for imperfect-information game solving. In Appendix E, we describe their multi-player, general-sum counterparts. Each instance is identified by parameters $p$ and $r$, which denote, respectively, the number of players and the number of ranks in the deck of cards. For example, a three-player Kuhn game with rank four is denoted by Kuhn3-4, or K3-4. We use different tie-breaking rules for the Goofspiel instances (denoted by A, DA, DH, AL—see Appendix E). Moreover, R$p$-$d$ denotes a random game instance with $p$ players and depth of the game tree $d$.

**Convergence** We evaluate the run time required by the algorithms to find an approximate CCE. The results are provided in Table 1, which reports the run time needed by CFR-S and CFR-Jr to achieve solutions with different levels of accuracy, and the time needed by CG for reaching an equilibrium. [7] The accuracy $\alpha$ of the $\varepsilon$-CCEs reached is defined as $\alpha = \frac{\varepsilon}{\Delta}$. Both CFR-S and CFR-Jr consistently outperform CG, as the latter fails to find a CCE in all instances except for the smallest ones (with less than 100 infosets). We also compare the convergence rates of CFR-S and CFR-Jr to that of CFR in multi-player, general-sum game instances. As expected, our experiments show that, in many instances, CFR fails to converge to a CCE. For instance, Figure 3, on the left, shows the performance of CFR-Jr, CFR-S (mean plus/minus standard deviation), and CFR over G2-4-DA in terms of accuracy $\alpha$. CFR performs dramatically worse than CFR-S and CFR-Jr, and it exhibits a non-convergent behavior with $\alpha$ being stuck above $4 \cdot 10^{-2}$.

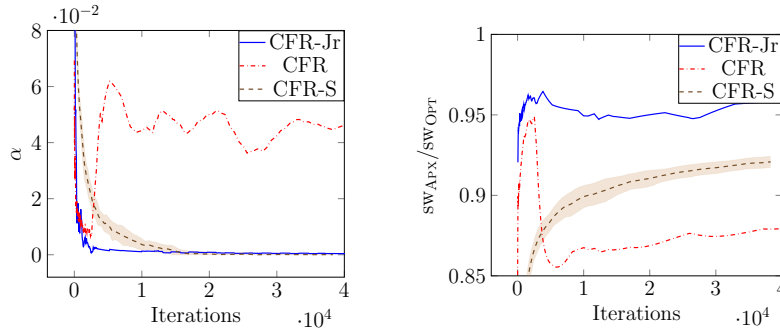

Figure 3: *Left*: Convergence rate attained in G2-4-DA. *Right*: Social welfare attained in G2-4-DA.

**Social welfare**   Table 1 shows, for the general-sum games, the social welfare approximation ratio between the social welfare of the solutions returned by the algorithms ($sw_{APX}$) and an upper bound on the optimal social welfare ($sw_{OPT}$). In particular, $sw_{OPT}$ is the maximum sum of players' utilities, which, while it is not guaranteed to be achievable by a CCE, it is always greater than or equal to the social welfare of an optimal CCE. [8] Interestingly, the approximation ratio provided by CFR-Jr is always better than that of CFR-S. Moreover, the social welfare guaranteed by CFR-Jr is always nearly optimal, which makes it a good heuristic to compute optimal CCEs. Reaching a socially good equilibrium is crucial, in practice, to make correlation credible. Figure 3, on the right, details the performance of CFR-Jr, CFR-S (mean plus/minus standard deviation), and CFR over G2-4-DA in terms of social welfare approximation ratio. CFR performs worse than the other two algorithms. This shows that, not only CFR does not converge to a CCE, but it is also not a good heuristic to find social-welfare-maximizing equilibria in multi-player, general-sum games.

## 7   Conclusions and future works

In this paper, we proved that finding an optimal (*i.e.*, social-welfare maximizing) CCE is not in Poly-APX, unless P = NP, in general-sum EFGs with two players and chance or with multiple players. We proposed CFR-Jr as an appealing remedy to the conundrum of computing correlated strategies for multi-player, general-sum settings with game instances beyond toy-problems. In the future, it would be interesting to further study how to approximate CCEs in other classes of structured games such as, *e.g.*, polymatrix games and congestion games. Moreover, a CCE strategy profile could be employed as a starting point to approximate tighter solution concepts which admit some form of correlation. This could be the case, *e.g.*, of the TMECor [14], which is used to model collusive behaviors and interactions involving teams. Finally, it would be interesting to further investigate whether it is possible to define regret-minimizing procedures for general EFGs leading to refinements of the CCEs, such as CEs and EFCEs. This begets new challenging problems in the study of how to minimize regret in structured games.

### Acknowledgments

We would like to thank Gabriele Farina for his helpful feedback. This work has been partially supported by the Italian MIUR PRIN 2017 Project ALGADIMAR "Algorithms, Games, and Digital Markets".

## Footnotes

[2]Formally, an $r$-approximation algorithm $\mathcal{A}$ for a maximization problem is such that $\frac{\text{OPT}}{\text{APX}} \leq r$, where OPT is the value of an optimal solution to the problem instance and APX is the value of the solution returned by $\mathcal{A}$. See [2] for additional details on approximation algorithms.

[3]Poly-APX is the class of optimization problems admitting a polynomial-time $\mathsf{poly}(\eta)$-approximation algorithm, where $\mathsf{poly}(\eta)$ is a polynomial function of the input size $\eta$ [2].

[4] Vector $\omega$ is a *realization-form strategy*, as defined by Farina et al. [14, Definition 2].

[5] Given a normal-form strategy $x_i \in \mathcal{X}_i$, its *support* is defined as the set of $\sigma_i \in \Sigma_i$ such that $x_i(\sigma_i) > 0$.

[6] The only other known algorithm to compute a CCE is by Huang and von Stengel [24] (see also [26] for an amended version). However, this algorithm relies on the ellipsoid method, which is inefficient in practice [20].

[7] Table 1 only accounts for algorithms with guaranteed convergence to a CCE (recall that CFR is not guaranteed to converge in multi-player, general-sum EFGs). The original version of the CG algorithm computes an optimal CCE. For our tests, we modified it to stop when a feasible solution is reached.

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
