[Supplementary Material]

## A    Discussion of other solution concepts based on correlation

The classical notion of correlated equilibrium is the one introduced by Aumann [1] for normal-form games. Its definition for EFGs is as follows, and it employs the equivalent normal form of the game.

**Definition 3.** *A* correlated equilibrium (CE) *of an EFG is a probability distribution $x^* \in \mathcal{X}$ such that, for every $i \in \mathcal{P}$, and for every $\sigma_i, \sigma_i' \in \Sigma_i$, it holds:*

$$\sum_{\sigma_{-i} \in \Sigma_{-i}} x^*(\sigma_i, \sigma_{-i}) \left( u_i(\sigma_i, \sigma_{-i}) - u_i(\sigma_i', \sigma_{-i}) \right) \geq 0.$$

A CE can be interpreted in terms of a mediator who, *ex ante* the play, draws the joint normal-form plan $\sigma^* \in \Sigma$ according to a publicly known $x^* \in \mathcal{X}$, and privately communicates each *recommendation* $\sigma_i^*$ to the corresponding player. After observing their recommended plan, each player decides whether to follow it or not.

CCEs (Definition 1) differ from CEs in that a CCE only requires that following the suggested plan is a best response in expectation, before the recommended plan is actually revealed. In both CE and CCE, the entire vector of recommendations $\sigma^* = (\sigma_i^*)_{i \in \mathcal{P}}$, specifying a move for each infoset, is computed before the playing phase of the game (as opposed to other solution concepts involving communication [16, 29]).

von Stengel and Forges [40] introduced the notion of *extensive-form correlated equilibrium* (EFCE). In this solution concept, each recommended action is assumed to be in a sealed envelope and is revealed only when the player reaches the relevant infoset (*i.e.*, the infoset where she can make that move). Therefore, EFCEs require recommendations to be delivered during game execution, which makes them more demanding in terms of communication requirements than CEs and CCEs. The size of the signal that has to be sampled is the same in all the three solution concepts, and it has polynomial size (one action for each infoset). The following relation holds between the sets of equilibria described above: CE $\subseteq$ EFCE $\subseteq$ CCE, see [40] for further details.

## B    Omitted proofs for inapproximability results

**Theorem 1.** *Given a two-player EFG with Nature, the problem of computing a social-welfare-maximizing CCE is not in Poly-APX unless* P = NP. [9]

*Proof.* We provide a reduction from SAT. Given a SAT instance $(C, V)$, we build a two-player EFG with Nature $\Gamma_\varepsilon(C, V)$ with the following structure:

- The game starts in $h^\varnothing \in I^\varnothing$, where player 1 chooses an action between $a^I$ and $a^O$. In the first case, the game goes on with $h^\varnothing \cdot a^I = h^N$. Otherwise, the game ends with $u_1(z) = 1$ and $u_2(z) = -1 + \varepsilon$.
- At state $h^N$, Nature selects an action among $\{a^\phi \mid \phi \in C\}$ uniformly at random, with $h^N \cdot a^\phi = h^\phi$.
- Each state $h^\phi$ constitutes a player 1's infoset $I^\phi$. At $I^\phi$, player 1 chooses an action in $\{a^{\phi,l} \mid l \in \phi\}$, where $l$ denotes a literal in $\phi$. Then, $h^\phi \cdot a^{\phi,l} = h^{\phi,l}$.
- All states $h^{\phi,l}$ such that $l = v$ or $l = \bar{v}$ for some $v \in V$ belong to the same player 2's infoset $I^v$. At $I^v$, player 2 has two actions available, namely $a^v$ and $a^{\bar{v}}$.
- Then, the game ends and players' payoffs $u_1(z) = u_2(z)$ are equal to 1 if and only if $z = h^{\phi,l} \cdot a^l$, while they are 0 otherwise.

Intuitively, each of the $2^{|V|}$ player 2's plans corresponds to a truth assignment $\tau$ where variable $v \in V$ is set to TRUE (resp., FALSE) if $a^v$ (resp., $a^{\bar{v}}$) is played at $I^v$. Moreover, a player 1's plan determines whether the game is played ($a^I$) or not ($a^O$) and, in the first case, it selects one literal for each clause $\phi \in C$ (corresponding to the action played at infoset $I^\phi$). If player 1 plays $a^I$, Nature chooses a clause $\phi \in C$ uniformly at random, and, then, the players' payoffs are 1 if and only if player 1 selected a literal of $\phi$ evaluating to TRUE under $\tau$. Thus, in this case, players' expected payoffs are equal to the number of literals selected by player 1 evaluating to TRUE under $\tau$, divided by the number of clauses

$|C|$. As a result, if SAT is satisfiable, then there exists a joint plan where players' expected payoffs are equal to 1. It is sufficient that player 2 plays the plan associated to a satisfying truth assignment $\tau$, while player 1 selects a literal evaluating to TRUE under $\tau$ for each clause. This is also a CCE with maximum social welfare equal to 2, as it provides the players with their maximum expected payoffs. Instead, if SAT is not satisfiable, then any CCE must recommend player 1 to play $a^O$ at $I^\varnothing$, otherwise her expected payoff would be strictly less than 1 and she would have an incentive to deviate to action $a^O$, reaching a payoff of 1. Hence, in this case, any CCE has social welfare $\varepsilon$. Now, let $\varepsilon = \frac{1}{2^\eta}$, where $\eta$ is the size of the SAT instance ($\varepsilon$ can be encoded with a number of bits polynomial in $|C|$ and $|V|$). Assume there is a polynomial-time poly$(\eta)$-approximation algorithm $\mathcal{A}$. If SAT is satisfiable, $\mathcal{A}$ applied to $\Gamma_\varepsilon(C, V)$ would return a CCE with social welfare at least $\frac{2}{\text{poly}(\eta)}$. Since, for $\eta$ sufficiently large, $\frac{2}{\text{poly}(\eta)} > \frac{1}{2^\eta}$, $\mathcal{A}$ would allow us to decide in polynomial time whether SAT is satisfiable, a contradiction unless P = NP. □

**Theorem 2.** *Given a three-player EFG without Nature, the problem of computing a social-welfare-maximizing CCE is not in Poly-*APX *unless* P = NP.

*Proof.* We use a reduction similar to that in Theorem 1. We build a three-player EFG $\hat{\Gamma}_\varepsilon(C, V)$ such that:

- The game starts in $h^\varnothing \in I^\varnothing$, as $\Gamma(C, V)$.
- At state $h^N$, player 3 plays an action $\{a^{O,\phi} \mid \phi \in C\} \cup \{a^I\}$, with $h^N \cdot a^I = h^I$, $h^N \cdot a^{O,\phi} = h^{O,\phi}$.
- All states $h^{O,\phi}$ and $h^I$ belong to a player 1's infoset $I^N$, where she selects an action among $\{a^{N,\phi} \mid \phi \in C\}$.
- Then, if player 3 played $a^I$, $h^I \cdot a^{N,\phi} = h^\phi$ and the games goes on as $\Gamma(C, V)$ (with player 3's payoffs set to zero). Instead, if player 3 played an action $a^{O,\phi}$, the game ends with $2u_1(z) = 2u_2(z) = -u_3(z) = \frac{|C|}{|C|-1}$ if $z = h^{O,\phi} \cdot a^{N,\phi'}$ and $\phi \neq \phi'$, while $2u_1(z) = 2u_2(z) = -u_3(z) = -|C|$ if $\phi = \phi'$.

Intuitively, the introduction of a third player allows us to simulate the random move of Nature in $\Gamma(C, V)$, since, in any CCE of $\hat{\Gamma}(C, V)$, player 1 is recommended to play a uniform distribution at infoset $I^N$ and player 3 is always told to play action $a^I$. First, if player 3 is recommended an action $a^{O,\phi}$ with positive probability, then player 2 would have an incentive to switch to action $a^O$ at $I^\varnothing$. Moreover, assuming player 3 is told to play $a^I$, if player 1 is recommended to play some action $a^{N,\phi}$ with probability $p > \frac{1}{|C|}$, then player 3 would have an incentive to switch to action $a^{O,\phi}$, as she would get $p|C| - (1-p)\frac{|C|}{|C|-1} > 0$, while she gets 0 by playing $a^I$. Finally, a reasoning similar to that for Theorem 1 concludes the proof. □

## C  CFR-S

### C.1  Example

Table 4 reports the game employed in the experiments of Figure 2, where the outer product of the average strategies $\bar{x}_1^T \otimes \bar{x}_2^T$ obtained by RM does not converge to a CCE as $T \to \infty$ (*i.e.*, for $\bar{x}_1^T \otimes \bar{x}_2^T$, $\varepsilon$ of the $\varepsilon$-CCE has a cyclic behavior and does not converge to zero).

| | | |
|---|---|---|
| 1,0 | 0,1 | 0,0 |
| 0,0 | 2,0 | 0,1 |
| 0,1 | 0,0 | 1,0 |

Figure 4: A simple variation of the Shapley game.

## C.2 Omitted proofs

The theoretical guarantees of CFR-S can be derived via the framework of Farina et al. [15], as discussed in the following.

At each iteration $t$, let $\sigma_i^t \in \Sigma_i$ be the normal-form plan sampled by player $i$ and $\sigma_{-i}^t \in \Sigma_{-i}$ be the plans drawn by the other players. The utility experienced by player $i$ at stage $t$ is denoted by $u_i^t(\sigma_i^t) := u_i(\sigma_i^t, \sigma_{-i}^t)$. Players' observations in CFR-S call for a slight variation in the definition of cumulative regret. After $T$ iterations, we define the cumulative regret experienced by player $i$ as

$$\tilde{R}_i^T := \max_{\hat{\sigma}_i \in \Sigma_i} \sum_{t=1}^{T} \left( u_i^t(\hat{\sigma}_i) - u_i^t(\sigma_i^t) \right). \tag{2}$$

The connection between the cumulative regret and the set of CCEs remains unchanged when the regret is defined as in Equation (2), as shown by the following result (whose proof is similar to that of [22, Proposition in Section 3]).

**Theorem 6.** *If* $\limsup_{T \to \infty} \frac{1}{T} \tilde{R}_i^T \leq 0$ *almost surely for each player* $i \in \mathcal{P}$*, then the empirical frequency of play* $\bar{x}^T$ *converges almost surely as* $T \to \infty$ *to the set of CCEs.*

*Proof.* By definition of cumulative regret, and by taking its average, we have

$$\limsup_{T \to \infty} \frac{1}{T} \max_{\hat{\sigma}_i \in \Sigma_i} \sum_{t=1}^{T} \left( u_i^t(\hat{\sigma}_i) - u_i^t(\sigma_i^t) \right) \leq 0,$$

which holds almost surely. Let $\sigma^t := (\sigma_i^t, \sigma_{-i}^t)$. It follows that, for each normal-form plan $\hat{\sigma}_i \in \Sigma_i$ we have

$$\frac{1}{T} \sum_{t=1}^{T} \left( u_i(\hat{\sigma}_i, \sigma_{-i}^t) - u_i(\sigma^t) \right) = \sum_{\sigma \in \Sigma} \bar{x}^T(\sigma) \left( u_i(\hat{\sigma}_i, \sigma_{-i}) - u_i(\sigma) \right).$$

Where $\bar{x}^T(\sigma)$ is the empirical frequency of $\sigma$ after $T$ iterations. On any subsequence where $\bar{x}^T$ converges, that is $\bar{x}^T \to x^* \in \mathcal{X}$, it holds almost surely, for each $\hat{\sigma}_i \in \Sigma_i$ that

$$\sum_{\sigma \in \Sigma} \bar{x}^T(\sigma) \left( u_i(\hat{\sigma}_i, \sigma_{-i}) - u_i(\sigma) \right) \to \sum_{\sigma \in \Sigma} x^*(\sigma) \left( u_i(\hat{\sigma}_i, \sigma_{-i}) - u_i(\sigma) \right).$$

The result immediately holds for Definition 1. □

In the following, we follow the approach of Farina et al. [15] to show how to decompose $\tilde{R}_i^T$ into regret terms which are computed locally at player $i$'s infosets. This allows us to avoid working with the (exponential-sized) normal form of an EFG even if $\tilde{R}_i^T$ is defined over player $i$'s normal-form plans. $\tilde{R}_i^T$ can be minimized via the minimization of other suitably defined regrets computed locally at player $i$'s infosets. In order to do this, we use the idea of *laminar regret decomposition* [15], but reasoning only on vertices of $\mathcal{X}_i$.

Given $\sigma_i \in \Sigma_i$, we denote by $\sigma_i(I)$ the action selected in $\sigma_i$ at infoset $I \in \mathcal{I}_i$. Moreover, $\sigma_{i \downarrow I}$ is the (sub)vector containing the actions selected in $\sigma_i$ at $I \in \mathcal{I}_i$ and all its descendant infosets.

First, we denote with $u_{i,I}^t : A(I) \to \mathbb{R}$ the *immediate utility* observed by player $i$ at infoset $I \in \mathcal{I}_i$, during iteration $t$. For every $a \in A(I)$, $u_{i,I}^t(a)$ is the utility experienced by player $i$ if the game ends after playing $a$ at $I$, without passing through another player $i$'s infoset.

Then, the following is player $i$'s utility attainable at infoset $I \in \mathcal{I}_i$ when a normal-form plan $\hat{\sigma}_i \in \Sigma_i$ is selected:

$$\hat{V}_I^t(\hat{\sigma}_{i \downarrow I}) := u_{i,I}^t(\hat{\sigma}_{i \downarrow I}(I)) + \sum_{I' \in \mathcal{C}_{I, \hat{\sigma}_{i \downarrow I}(I)}} \hat{V}_{I'}^t(\hat{\sigma}_{i \downarrow I'}), \tag{3}$$

where $\mathcal{C}_{I,a} \subseteq \mathcal{I}_i$ is the set of possible next player $i$'s infosets, given that she played action $a \in A(I)$ at infoset $I \in \mathcal{I}_i$. We introduce a parameterized utility function, which is used to define regrets locally at each infoset, and reads as follows:

$$\hat{u}_{i,I}^t : a \in A(I) \mapsto u_{i,I}^t(a) + \sum_{I' \in \mathcal{C}_{I,a}} \hat{V}_{I'}^t(\sigma_{i \downarrow I'}^t). \tag{4}$$

The utility function $\hat{u}_{i,I}^t$ preserves convexity of $u_i^t$. Finally, we modify the notion of *laminar regret*, as

$$\hat{R}_I^t := \max_{a \in A(I)} \sum_{t=1}^{T} \hat{u}_{i,I}^t(a) - \sum_{t=1}^{T} \hat{u}_{i,I}^t(\sigma_i^t(I)). \tag{5}$$

Let $V_I^t := \hat{V}_I^t(\sigma_{i,\downarrow I}^t)$. Then, we introduce the cumulative regret at infoset $I \in \mathcal{I}_i$, defined as

$$R_{\downarrow I}^T := \max_{\hat{\sigma}_{i\downarrow I}} \sum_{t=1}^{T} \hat{V}_I^t(\hat{\sigma}_{i\downarrow I}) - \sum_{t=1}^{T} V_I^t. \tag{6}$$

**Lemma 7.** *The cumulative regret at each infoset $I \in \mathcal{I}_i$ can be decomposed as*

$$R_{\downarrow I}^T = \max_{a \in A(I)} \left( \sum_{t=1}^{T} \hat{u}_{i,I}^t(a) + \sum_{I' \in \mathcal{C}_{I,a}} R_{\downarrow I'}^T \right) - \sum_{t=1}^{T} \hat{u}_{i,I}^t(\sigma_i^t(I)).$$

*Proof.* By definition of cumulative regret at $I \in \mathcal{I}_i$ we have that:

$$R_{\downarrow I}^T := \max_{\hat{\sigma}_{i\downarrow I}} \sum_{t=1}^{T} \hat{V}_I^t(\hat{\sigma}_{i\downarrow I}) - \sum_{t=1}^{T} V_I^t =$$

$$= \max_{\hat{\sigma}_{i\downarrow I}} \sum_{t=1}^{T} \left( u_{i,I}^t(\hat{\sigma}_{i\downarrow I}(I)) + \sum_{I' \in \mathcal{C}_{I,\hat{\sigma}_{i\downarrow I}^t(I)}} \hat{V}_{I'}^t(\hat{\sigma}_{i\downarrow I'}) \right) - \sum_{t=1}^{T} V_I^t =$$

$$= \max_{a \in A(I)} \left( \sum_{t=1}^{T} u_{i,I}^t(a) + \sum_{I' \in \mathcal{C}_{I,a}} \max_{\hat{\sigma}_{i\downarrow I'}} \sum_{t=1}^{T} \hat{V}_{I'}^t(\hat{\sigma}_{i\downarrow I'}) \right) - \sum_{t=1}^{T} V_I^t.$$

Then, by employing Equation (6), we get

$$R_{\downarrow I}^T = \max_{a \in A(I)} \left( \sum_{t=1}^{T} u_{i,I}^t(a) + \sum_{I' \in \mathcal{C}_{I,a}} \left( R_{\downarrow I'}^T + \sum_{t=1}^{T} V_{I'}^t \right) \right) - \sum_{t=1}^{T} V_I^t.$$

Finally, we obtain the result by rewriting terms according to Equation (4). $\square$

The following theorem shows that, in order to minimize $\tilde{R}_i^T$, it is enough to minimize the laminar regret locally at each $I \in \mathcal{I}_i$ as defined in Equation (5).

**Lemma 8.** *The cumulative regret $\tilde{R}_i^T$ satisfies the following:*

$$\tilde{R}_i^T \leq \max_{\hat{\sigma}_i \in \Sigma_i} \sum_{I \in \mathcal{I}_i} \rho_I^{\hat{\sigma}_i} \hat{R}_I^T.$$

*Proof.* Consider a generic infoset $I \in \mathcal{I}_i$. By exploiting Lemma 7 and Definition 5, we can write:

$$R_{\downarrow I}^T = \max_{a \in A(I)} \left( \sum_{t=1}^{T} \hat{u}_{i,I}^t(a) + \sum_{I' \in \mathcal{C}_{I,a}} R_{\downarrow I'}^T \right) - \sum_{t=1}^{T} \hat{u}_{i,I}^t(\sigma_i^t(I)) \leq$$

$$\leq \max_{a \in A(I)} \sum_{t=1}^{T} \hat{u}_{i,I}^t(a) + \max_{a \in A(I)} \sum_{I' \in \mathcal{C}_{I,a}} R_{\downarrow I'}^T - \sum_{t=1}^{T} \hat{u}_{i,I}^t(\sigma_i^t(I)) =$$

$$= \hat{R}_I^T + \max_{a \in A(I)} \sum_{I' \in \mathcal{C}_{I,a}} R_{\downarrow I'}^T.$$

By starting from the root of the game and applying the above equation inductively, we obtain our result. $\square$

The last result provides an immediate proof of the following.

**Theorem 3.** *The empirical frequency of play $\bar{x}^T$ obtained with CFR-S converges to a CCE almost surely, for $T \to \infty$.*

*Proof.* CFR-S minimizes each laminar regret $\hat{R}_I^T$, as defined in Equation (5), through standard RM, which guarantees that $\limsup_{T \to \infty} \frac{1}{T} \hat{R}_I^T \leq 0$ almost surely. Therefore, $\limsup_{T \to \infty} \frac{1}{T} \tilde{R}_i^T \leq 0$ almost surely (Lemma 8), which implies that the empirical frequency of play converges almost surely to a CCE for $T \to \infty$ (Theorem 6). □

Finally, we observe that, at each iteration $t$ and infoset $I \in \mathcal{I}_i$, $\sigma_i^t(I)$ is selected according to the strategy $\pi_{i,I}^t$ recommended by the regret minimizer at infoset $I$. Thus, $\sigma_i^t$ is drawn with probability $\prod_{I \in \mathcal{I}_i} \pi_{i,I,\sigma_i^t(I)}^t$, which is equal to $x_i^t(\sigma_i^t)$, where $x_i^t \in \mathcal{X}_i$ is the normal-form strategy realization equivalent to the behavioral strategy $\pi_i^t$.

# D  CFR-Jr

## D.1  Algorithm pseudocode

For the sake of completeness and clarity, in Algorithm 3 we provide the pseudocode of the CFR-Jr algorithm, which uses a vanilla implementation of the CFR algorithm as a subroutine.

---

**Algorithm 3** `CFR-Jr`

---

1:  **function** CFR-Jr($\Gamma$)
2:      Initialize the joint strategy $\bar{x}$ to all zeros
3:      $t \leftarrow 0$
4:      **while** $t < T$ **do**
5:          **for all** $i \in \mathcal{P}$ **do**
6:              $\pi_i^t \leftarrow \text{CFR}(\Gamma, i)$
7:              $x_i^t \leftarrow \text{NF-Strategy-Reconstruction}(\pi_i^t)$
8:          $\bar{x} \leftarrow \bar{x} + \bigotimes_{i \in \mathcal{P}} x_i^t$  $\quad \triangleright \bigotimes_{i \in \mathcal{P}} x_i^t$ is joint distribution $x^t$ defined as the product of the players' normal-form strategies
9:          $t \leftarrow t + 1$
10:     **return** $\bar{x}^T = \bar{x}/T$

---

CFR-Jr maintains a variable $\bar{x}$ which stores the sum of the joint probability distributions $x^t$ (notice that it may be compactly represented using a dictionary, as for $x_i$ in Algorithm 2). CFR-Jr executes, at each $t$, an iteration of the CFR algorithm (Line 6). In particular, the CFR subroutine executes a step of vanilla CFR, including the update of regrets and behavioral strategies. In addition, at each iteration $t$, CFR-Jr constructs normal-form strategies $x_i^t$ (one per player $i \in \mathcal{P}$) which are realization equivalent to the behavioral strategies $\pi_i^t$ obtained via CFR (Line 7). Then the product $x^t$ of the players' normal-strategies is computed and added to $\bar{x}$ (Line 8). Notice that $\bar{x}$ is not used by the CFR subroutine to update the players' strategies and regrets. Finally, CFR-Jr returns the $\bar{x}$ divided by $T$, which is the average $\bar{x}^T$.

## D.2  Omitted proofs

In order to give the complete proof of Theorem 4, we first need to prove two lemmas concerning the existence of a normal-form plan $\bar{\sigma}_i$ such that $\bar{\omega} = \min_{z \in Z(\bar{\sigma}_i)} \omega_z > 0$ whenever the vector $\omega$ has at least one strictly positive component.

To simplify the presentation, we introduce some additional notation. Extending the definition of $Z(\sigma_i)$, let $Z(I, a)$ be the set of terminal nodes potentially reachable from infoset $I \in \mathcal{I}_i$ when player $i$ selects $a \in A(I)$. Moreover, we denote by $Z(\sigma_i, I, a)$ the set of terminal nodes potentially reachable from $I$ after playing action $a$ at $I$, and then following the actions prescribed by $\sigma_i \in \Sigma_i$. $Z(I)$ and $Z(\sigma_i, I)$ are defined analogously.

Observe that, in Line 5 of Algorithm 2, the normal-form plan $\bar{\sigma}_i \in \arg\max_{\sigma_i \in \Sigma_i} \min_{z \in Z(\sigma_i)} \omega_z$ can be recursively built, while traversing the game tree. Let $\Sigma_i^\omega$ be a subset of $\Sigma_i$ recursively defined as follows:

$$\Sigma_i^\omega := \left\{ \bar{\sigma}_i \in \Sigma_i | \forall I \in \mathcal{I}_i, \bar{\sigma}_i(I) \in \arg\max_{a \in A(I)} \min_{z \in Z(\bar{\sigma}_i, I, a)} \omega_z \right\}. \tag{7}$$

Any $\bar{\sigma}_i \in \Sigma_i^\omega$ is a feasible result of Line 5 in Algorithm 2.

**Lemma 9.** *Given $\bar{\sigma}_i \in \Sigma_i^\omega$, it holds that*

$$\max_{a \in A(I)} \min_{z \in Z(\bar{\sigma}_i, I, a)} \omega_z = 0 \qquad \forall I \in \mathcal{I}_i,$$

*if and only if $\omega = \mathbf{0}$.* [10]

*Proof.* The proof is by induction on the depth of the game tree. Let $\mathcal{C}_{I,a}$ be the set of player $i$'s infosets immediately reachable by playing action $a \in A(I)$ at infoset $I \in \mathcal{I}_i$.

As for the base case of the induction, let us consider $I \in \mathcal{I}_i$ such that $\mathcal{C}_{I,a} = \emptyset$ for all $a \in A(I)$. By the definition of $\omega$ we have: $\omega_z = \omega_{z'} = \rho_I^{\pi_i} \pi_{i,I,a}$, for each $a \in A(I)$, and each pair $z, z' \in Z(I, a)$. This implies

$$\max_{a \in A(I)} \min_{z \in Z(\bar{\sigma}_i, I, a)} \omega_z = \max_{a \in A(I)} \rho_I^{\pi_i} \pi_{i,I,a} = \max_{z \in Z(I)} \omega_z.$$

Clearly, the max of a non-negative function over a set is zero iff the function is zero for all the elements of the set. Then,

$$\max_{a \in A(I)} \min_{z \in Z(\bar{\sigma}_i, I, a)} \omega_z = \max_{z \in Z(I)} \omega_z = 0$$

iff $\omega_z = 0$ for all $z \in Z(I)$.

As for the inductive step, let us consider a generic infoset $I \in \mathcal{I}_i$. It holds that $Z(\bar{\sigma}_i, I, \bar{a}) = Z(\bar{\sigma}_i, I)$ if

$$\bar{a} \in \arg\max_{a \in A(I)} \min_{z \in Z(\bar{\sigma}_i, I, a)} \omega_z. \tag{8}$$

By reasoning as above, we can conclude that $\max_{a \in A(I)} \min_{z \in Z(\bar{\sigma}_i, I, a)} \omega_z = 0$ iff $\min_{z \in Z(\bar{\sigma}_i, I, a)} \omega_z = 0$ for all $a \in A(I)$. Now, take any pair $(a', I') \in A(I) \times \mathcal{C}_{I,a'}$, by applying the above observation it follows that

$$\max_{a \in A(I')} \min_{z \in Z(\bar{\sigma}_i, I', a)} \omega_z = \min_{z \in Z(\bar{\sigma}_i, I', \bar{a})} \omega_z = \min_{z \in Z(\bar{\sigma}_i, I')} \omega_z,$$

where $\bar{a}$ is computed as in (8). Being $I'$ a descendant of $I$, we have that $Z(I') \subseteq Z(I, a')$ and, in particular, $Z(\bar{\sigma}_i, I') \subseteq Z(\bar{\sigma}_i, I, a')$. Thus, $\min_{z \in Z(\bar{\sigma}_i, I, a')} \omega_z = 0$ implies that $\min_{z \in Z(\bar{\sigma}_i, I')} \omega_z = 0$. By the induction hypothesis, we have that for $I' \in \mathcal{I}_i$ following $I \in \mathcal{I}_i$, it holds

$$\max_{a \in A(I')} \min_{z \in Z(\bar{\sigma}_i, I', a)} \omega_z = 0$$

iff $\omega_z = 0$ for every $z \in Z(I')$. Then,

$$\max_{a \in A(I)} \min_{z \in Z(\bar{\sigma}_i, I, a)} \omega_z = 0 \quad \Leftrightarrow \quad \min_{z \in Z(\bar{\sigma}_i, I, a)} \omega_z = 0 \ \forall a \in A(I),$$

where the right term is true iff

$$\max_{a \in A(I')} \min_{z \in Z(\bar{\sigma}_i, I', a)} \omega_z = 0 \ \forall a' \in A(I), I' \in \mathcal{C}_{I,a'}.$$

This holds, by inductive hypothesis, iff

$$\omega_z = 0 \ \forall z \in Z(I'), \quad \forall a' \in A(I), I' \in \mathcal{C}_{I,a'}.$$

Given that $Z(I) = \bigcup_{a' \in A(I), I' \in \mathcal{C}_{I,a'}} Z(I')$, the last condition holds iff $\omega_z = 0$ for all $z \in Z(I)$, which concludes the proof. $\square$

**Lemma 10.** *If $\omega > 0$, then a normal-form plan $\bar{\sigma}_i \in \Sigma_i^\omega$ is such that $\min_{z \in Z(\bar{\sigma}_i)} \omega_z > 0$.*

*Proof.* Let $I^\varnothing$ be the root infoset of player $i$. Observe that $Z(\bar{\sigma}_i) = Z(\bar{\sigma}_i, I^\varnothing, \bar{a})$ if $\bar{a} \in \arg\max_{a \in A(I^\varnothing)} \min_{z \in Z(\bar{\sigma}_i, I^\varnothing, a)} \omega_z$. Applying Lemma 9 to infoset $I^\varnothing$, we have that

$$\max_{a \in A(I^\varnothing)} \min_{z \in Z(\bar{\sigma}_i, I^\varnothing, a)} \omega_z = \min_{z \in Z(\bar{\sigma}_i)} \omega_z = 0$$

iff $\omega_z = 0$ for every $z \in Z(I^\varnothing) = Z$. Since $\omega_z \geq 0$ for all $z \in Z$, we have $\min_{z \in Z(\bar{\sigma}_i)} \omega_z > 0$ iff $\omega > \mathbf{0}$. Being this last condition always verified within the main loop of Algorithm 2, we have that a normal-form plan $\bar{\sigma}_i \in \arg\max_{\sigma_i \in \Sigma_i} \min_{z \in Z(\sigma_i)} \omega_z$ is such that $\min_{z \in Z(\bar{\sigma}_i)} \omega_z > 0$. $\qquad\square$

**Theorem 4.** *Algorithm 2 outputs a normal-form strategy $x_i \in \mathcal{X}_i$ realization equivalent to a given behavioral strategy $\pi_i$, and it runs in time $O(|Z|^2)$. Moreover, $x_i$ has support size at most $|Z|$.*

*Proof.* **Time complexity**. Given $\bar{\sigma}_i$, for each $z \in Z(\bar{\sigma}_i)$ it holds $\bar{\omega} \rho_z^{\bar{\sigma}_i} = \bar{\omega} = \min_{z \in Z(\bar{\sigma}_i)} \omega_z$, and, for each $z \notin Z(\bar{\sigma}_i)$, $\bar{\omega} \rho_z^{\bar{\sigma}_i} = 0$. Then, after each iteration, we have $\omega \geq \mathbf{0}$. Moreover, it holds $\omega_{\bar{z}} = 0$ for each $\bar{z} \in \arg\min_{z \in Z(\bar{\sigma}_i)} \omega_z$. Then, at each iteration, at least one component of $\omega$ goes from being $> 0$, to 0. Given that $\omega$ is always non-negative, we have that the vector $\omega$ is zeroed in at most $|Z|$ iterations. Each iteration runs in $O(\max\{|\mathcal{I}_i|, |Z|\})$, as $\bar{\sigma}_i$ can be recursively computed by iterating on the infosets in a bottom-up fashion, while each $\omega$ update needs to consider each terminal node at most once. Given that for each non-degenerate game tree (*i.e.*, $A(I) > 1$ for all $I \in \mathcal{I}_i$) we have $|\mathcal{I}_i| \leq |Z|$, the overall complexity of the algorithm is $O(|Z|^2)$.

**Support size**. No normal-form plan can be selected more than once because: i) after $\bar{\sigma}_i$ is selected, at least one component of $\omega$ in $Z(\bar{\sigma}_i)$ is zeroed; ii) $\bar{\sigma}_i$ is selected so that $\min_{z \in Z(\bar{\sigma}_i)} \omega_z > 0$. Then, the support of $x_i$ has size equal to the number of normal-form plans $\bar{\sigma}_i$ selected at each iteration of Algorithm 2, which is at most $|Z|$.

**Realization equivalence**. Let $\bar{\sigma}_i^k$ be the normal-form plan selected at the $k$-th iteration. By recursively expanding

$$\omega_z \leftarrow \omega_z - \bar{\omega}\, \rho_z^{\bar{\sigma}_i}$$

we obtain the following (for clarity, we add apices indicating the iteration):

$$\omega_z^k = \omega_z^{k-1} - \rho_z^{\bar{\sigma}_i^k} \min_{z' \in Z(\bar{\sigma}_i^{k-1})} \omega_{z'}^{k-1} =$$

$$= \omega_z^{k-2} - \rho_z^{\bar{\sigma}_i^{k-1}} \min_{z' \in Z(\bar{\sigma}_i^{k-2})} \omega_{z'}^{k-2} - \rho_z^{\bar{\sigma}_i^k} \min_{z' \in Z(\bar{\sigma}_i^{k-1})} \omega_{z'}^{k-1} =$$

$$= \ldots = \omega_z^0 - \sum_{k'=1}^{k} \rho_z^{\bar{\sigma}_i^{k'}} \min_{z' \in Z(\bar{\sigma}_i^{k'-1})} \omega_{z'}^{k'-1}.$$

Suppose that the algorithm alts at iteration $k$. Then $\omega^k = \mathbf{0}$, which gives:

$$\omega_z^0 = \sum_{k'=1}^{k} \rho_z^{\bar{\sigma}_i^{k'}} \min_{z' \in Z(\bar{\sigma}_i^{k'-1})} \omega_{z'}^{k'-1}.$$

Finally, we show that $x_i$ and $\pi_i$ are realization equivalent by checking that they force the same distribution over $Z$. We have, for each $z \in Z$:

$$\rho_z^{x_i} = \sum_{\sigma_i \in \Sigma_i} \rho_z^{\sigma_i} x_i(\sigma_i) =$$

$$= \sum_{\sigma_i \in \{\bar{\sigma}_i^{k'}\}_{k'=1}^{k}} \rho_z^{\sigma_i} x_i(\sigma_i) = \sum_{k'=1}^{k} \rho_z^{\bar{\sigma}_i^{k'}} x_i(\bar{\sigma}_i^{k'}) =$$

$$= \sum_{k'=1}^{k} \rho_z^{\bar{\sigma}_i^{k'}} \min_{z' \in Z(\bar{\sigma}_i^{k'-1})} \omega_{z'}^{k'-1} = \rho_z^{\pi_i},$$

where $\pi_i$ is the behavioral strategy given in input. This concludes the proof. $\qquad\square$

**Theorem 5.** *If $\frac{1}{T}R_i^T \leq \varepsilon$ for each player $i \in \mathcal{P}$, then $\bar{x}^T$ obtained with CFR-Jr is an $\varepsilon$-CCE.*

*Proof.* First, let us recall that $x^t \in \mathcal{X}$ is defined in such a way that $x^t(\sigma) = \prod_{i \in \mathcal{P}} x_i^t(\sigma_i)$ for every joint normal-form plan $\sigma \in \Sigma$, with $\sigma = (\sigma_i)_{i \in \mathcal{P}}$. By assumption, $\frac{1}{T}R_i^T \leq \varepsilon$ implies the following:

$$\max_{\hat{\sigma}_i \in \Sigma_i} \left( \sum_{t=1}^{T} \sum_{\sigma_{-i} \in \Sigma_{-i}} u_i(\hat{\sigma}_i, \sigma_{-i}) \prod_{j \neq i \in \mathcal{P}} x_j^t(\sigma_j) + \right.$$
$$\left. - \sum_{t=1}^{T} \sum_{\sigma_i \in \Sigma_i} \sum_{\sigma_{-i} \in \Sigma_{-i}} u_i(\sigma_i, \sigma_{-i}) \prod_{j \in \mathcal{P}} x_j^t(\sigma_j) \right) \leq \varepsilon T.$$

Moreover, since the condition holds for every $i \in \mathcal{P}$, by re-writing the max operator we get, $\forall i \in \mathcal{P}, \hat{\sigma}_i \in \Sigma_i$:

$$\sum_{t=1}^{T} \sum_{\sigma_{-i} \in \Sigma_{-i}} u_i(\hat{\sigma}_i, \sigma_{-i}) \prod_{j \neq i} x_j^t(\sigma_j) \quad - \quad \sum_{t=1}^{T} \sum_{\sigma_i \in \Sigma_i} \sum_{\sigma_{-i} \in \Sigma_{-i}} u_i(\sigma_i, \sigma_{-i}) \prod_{j \in \mathcal{P}} x_j^t(\sigma_j) \quad \leq \quad \varepsilon T.$$

Since $\sum_{\sigma_i \in \Sigma_i} x_i^t(\sigma_i) = 1$, it follows that $\sum_{\sigma_{-i} \in \Sigma_{-i}} u_i(\hat{\sigma}_i, \sigma_{-i}) \prod_{j \neq i \in \mathcal{P}} x_j^t(\sigma_j)$ is equal to $\sum_{\sigma_i \in \Sigma_i} \sum_{\sigma_{-i} \in \Sigma_{-i}} u_i(\hat{\sigma}_i, \sigma_{-i}) \prod_{j \in \mathcal{P}} x_j^t(\sigma_j)$. Thus,

$$\sum_{t=1}^{T} \sum_{\sigma_i \in \Sigma_i} \sum_{\sigma_{-i} \in \Sigma_{-i}} \prod_{j \in \mathcal{P}} x_j^t(\sigma_j) \left( u_i(\hat{\sigma}_i, \sigma_{-i}) - u_i(\sigma_i, \sigma_{-i}) \right) \leq \varepsilon T \quad \forall i \in \mathcal{P}, \hat{\sigma}_i \in \Sigma_i.$$

Using the definition of $\bar{x}^T$, we obtain

$$\sum_{t=1}^{T} \sum_{\sigma_i \in \Sigma_i} \sum_{\sigma_{-i} \in \Sigma_{-i}} \bar{x}^T(\sigma_i, \sigma_{-i}) \left( u_i(\hat{\sigma}_i, \sigma_{-i}) - u_i(\sigma_i, \sigma_{-i}) \right) \leq \varepsilon \quad \forall i \in \mathcal{P}, \hat{\sigma}_i \in \Sigma_i,$$

which proves that $\bar{x}^T$ is an $\varepsilon$-CCE. $\qquad\square$

# E    Additional details on the experimental evaluation

In this section we provide further details on the experimental evaluation.

## E.1    Experimental setup

The multi-player games instances that we employ are structured as follows.

**Kuhn Poker**. In Kuhn3-$r$ (K3-$r$), each player initially pays one chip to the pot, and is dealt a single private card. Then, players act in turns. The first player may check or bet (*i.e.*, paying an additional chip to the pot). The second player can either fold/call the bet, or check/bet after an initial check of the first player. At this point, if no bets have been placed, the third player decides between checking or betting. Otherwise, she can either fold or call. If the third player bets, then the others have to choose between folding or calling. At the showdown, the player with the highest card who has not folded wins all the chips in the pot.

**Leduc Hold'em Poker**. We employ three-player variants larger than the two-player version usually employed [37]. In our enlarged variants, Leduc3-$r$ (L3-$r$) contains three suits and $r \geq 3$ card ranks (*i.e.*, it contains triples of cards $A, 2, \ldots, r$ for a total of $3\,r$ cards). Each player initially pays one chip to the pot, and is dealt a single private card. After a first round of betting (with betting parameter $k_1$), a community card is dealt face up. Then, a second round of betting is played (with betting parameter $k_2$). Finally, a showdown occurs and players that did not fold reveal their private cards. If a player pairs her card with the community card, she wins the pot. Otherwise, the player with the highest private card wins. In the event that all players have the same private card, they draw and split the pot. Betting rounds follow the same rules of Kuhn Poker. We set $k_1 = 2$ and $k_2 = 4$. These

are the numbers of chips that a player has to pay to bet/call in the first and second round of betting, respectively.

**Goofspiel.** In addition to Poker games, we consider the game of Goofspiel. In this game, cards rank A (low), 2, . . ., 10, J, Q, K (high). When scoring points, the Ace is worth 1 point, cards 2-10 their face value, Jack 11, Queen 12, and King 13. Goofspiel $p$-$r$ ($p$ is the number of players) employs $p + 1$ suits, each containing cards A, . . .,$r$. One suit is singled out as the prizes. The prizes are shuffled and placed between the players, with the top card turned face up. Each of the remaining suits becomes the hand of one of the players. The game proceeds in rounds. Each player selects a card from her hand, keeping her choice secret from the opponent. Once all players have selected a card, they are simultaneously revealed, and the player with the highest bid wins the prize card. We employ the following tie breaking rules to obtain different kinds of instances. Some of them (*e.g.*, *Accumulate*) are *almost* constant-sum games (*i.e.*, constant sum for all but few outcomes), while others (*e.g.*, *Discard always*) present larger differences in the sum of payoffs attainable at different terminal nodes:

- *Accumulate* (A): the prize card goes to the player that selected the highest unique card. If all players selected the same card, the prize card is taken aside and the game continues unveiling the next one: the winner (if any) of the new round will take both prize cards. The process is repeated until the tie is broken or the game ends, in which case all prize cards that have been taken aside are discarded.
- *Discard-if-all* (DA): the prize card goes to the player that selected the highest unique card; if all players selected the same card, the prize card is discarded.
- *Discard-if-high* (DH): if the tie is on the highest-valued card, then the prize card is discarded; otherwise, the prize card goes to the player that selected the highest unique card.
- *Discard always* (AL): the prize card is discarded and the game goes on with the next round.

The game ends when the players terminate their cards. Players calculate their final utility by summing up the value of the prize cards they won.

## E.2 Full experimental results

CFR, CFR-S and CFR-Jr algorithms have all been implemented in the Python 3 language, and CG employs the GUROBI 8.0 MILP solver to solve the pricing problems. All the experiments are run with a 24 hours time-limit on a UNIX machine with a total of 32 cores working at 2.3 GHz, equipped with 128 GB of RAM.

Tables 2 and 3 provide the complete comparison between CFR-S, CFR-Jr, and CG. We remark that the utility ratio $sw_{APX}/sw_{OPT}$ is computed only for general-sum instances, and that by >24h we mean that the execution was killed by the time limit imposed on the system.

| Game | Tree size #infosets | $\Delta$ | CFR-S | | | | | | | CG |
|---|---|---|---|---|---|---|---|---|---|---|
| | | | $\alpha = 0.1$ | $\alpha = 0.05$ | $\alpha = 0.01$ | $\alpha = 0.005$ | $\alpha = 0.001$ | $\alpha = 0.0005$ | $sw_{APX}/sw_{OPT}$ | |
| K3-6 | 72 | 6 | 0.22s | 1.41s | 24m20s | 9h15m | > 24h | > 24h | - | 3h47m |
| K3-7 | 84 | 6 | 0.62s | 4.22s | 1h3m | 17h11m | > 24h | > 24h | - | 14h37m |
| K3-10 | 120 | 6 | 1.89s | 22.69s | 11h19m | > 24h | > 24h | > 24h | - | > 24h |
| L3-4 | 1200 | 21 | 4.0s | 10m33s | > 24h | > 24h | > 24h | > 24h | - | > 24h |
| L3-6 | 2664 | 21 | 21.54s | 2h5m | > 24h | > 24h | > 24h | > 24h | - | > 24h |
| L3-8 | 4704 | 21 | 35.3s | 13h55m | > 24h | > 24h | > 24h | > 24h | - | > 24h |
| G2-4-A⋆ | 4856 | 10 | 1m11s | 10m31s | 27h3m | > 24h | > 24h | > 24h | 0.979 | > 24h |
| G2-4-DA⋆ | 4856 | 10 | 12.83s | 2m1s | 53m28s | 3h28m | 4h48m | 4h17m | 0.918 | > 24h |
| G2-4-DH⋆ | 4856 | 10 | 11.56s | 1m19s | 42m18s | 2h7m | 3h19m | 3h28m | 0.918 | > 24h |
| G2-4-AL⋆ | 4856 | 10 | 15.01s | 2m3s | 43m18s | 1h33m | 4h4m | 4h20m | 0.919 | > 24h |
| G3-4-A⋆ | 98508 | 10 | 6m19s | 1h33m | > 24h | > 24h | > 24h | > 24h | 0.995 | > 24h |
| G3-4-DA⋆ | 98508 | 10 | 9m17s | 1h13m | 17h12m | > 24h | > 24h | > 24h | 0.986 | > 24h |
| G3-4-DH⋆ | 98508 | 10 | 5m24s | 47m33s | 11h51m | 19h40m | 22h11m | > 24h | 0.886 | > 24h |
| G3-4-AL⋆ | 98508 | 10 | 2m23s | 32m34s | 10h25m | 15h32m | 14h36m | 17h30m | 0.692 | > 24h |
| R3-12⋆ | 3071 | 1 | 45.0s | 1m44s | 13m10s | 35m38s | 10h8m | 3h8m | 0.906 | > 24h |
| R3-15⋆ | 24542 | 1 | 10m5s | 21m30s | 2h5m | 4h28m | 3h25m | 7h50m | 0.924 | > 24h |

Table 2: Running times and social welfare obtained by the CFR-S algorithm (for various levels of accuracy), and the CG algorithm. General-sum instances are marked with ⋆.

| Game | Tree size #infosets | $\Delta$ | CFR-Jr | | | | | | | CG |
|---|---|---|---|---|---|---|---|---|---|---|
| | | | $\alpha=0.1$ | $\alpha=0.05$ | $\alpha=0.01$ | $\alpha=0.005$ | $\alpha=0.001$ | $\alpha=0.0005$ | $sw_{APX}/sw_{OPT}$ | |
| K3-6 | 72 | 6 | 1.03s | 1.03s | 4.55s | 13.41s | 1m7s | 11m21s | - | 3h47m |
| K3-7 | 84 | 6 | 2.35s | 2.35s | 7.92s | 14.33s | 10m49s | 51m27s | - | 14h37m |
| K3-10 | 120 | 6 | 7.21s | 7.21s | 17.2s | 72.78s | 31m41s | 4h11m | - | > 24h |
| L3-4 | 1200 | 21 | 1.72s | 1m15s | 1h1m | 6h10m | > 24h | > 24h | - | > 24h |
| L3-6 | 2664 | 21 | 8.2s | 2m40s | 2h35m | 1h19m | > 24h | > 24h | - | > 24h |
| L3-8 | 4704 | 21 | 7m44s | 20m22s | 17h32m | > 24h | > 24h | > 24h | - | > 24h |
| G2-4-A$^\star$ | 4856 | 10 | 5m28s | 20m23s | 4h3m | 11h4m | > 24h | > 24h | 0.994 | > 24h |
| G2-4-DA$^\star$ | 4856 | 10 | 1m3s | 1m36s | 14m31s | 56m6s | > 24h | > 24h | 0.976 | > 24h |
| G2-4-DH$^\star$ | 4856 | 10 | 1m10s | 1m51s | 16m27s | 1h5m | > 24h | > 24h | 0.976 | > 24h |
| G2-4-AL$^\star$ | 4856 | 10 | 1m10s | 1m48s | 15m2s | 55m43s | > 24h | > 24h | 0.976 | > 24h |
| G3-4-A$^\star$ | 98508 | 10 | 1h21s | 1h3m | 3h3m | 4h13m | 5h4m | > 24h | 0.999 | > 24h |
| G3-4-DA$^\star$ | 98508 | 10 | 9m25s | 12m18s | 1h1m | 1h50m | > 24h | > 24h | 1.000 | > 24h |
| G3-4-DH$^\star$ | 98508 | 10 | 13m59s | 16m38s | 2h21m | 4h8m | 8h50m | 15h27m | 1.000 | > 24h |
| G3-4-AL$^\star$ | 98508 | 10 | 13m55s | 1h21m | 1h38m | 5m2s | > 24h | > 24h | 0.730 | > 24h |
| R3-12$^\star$ | 3052 | 1 | 7.67s | 16.94s | 1m37s | 3m19s | 17m1s | 24m6s | 0.897 | > 24h |
| R3-15$^\star$ | 24588 | 1 | 3m10s | 3m34s | 9m1s | 14m53s | 1h19m | 3h3m | 0.931 | > 24h |

Table 3: Running times and social welfare obtained by the CFR-Jr algorithm (for various levels of accuracy), and the CG algorithm. General-sum instances are marked with $^\star$.

We tested CFR-Jr also on some extensive-form variants of the Shapley game [35], a normal-form general-sum 3x3 game that has been shown to induce cyclic, non-convergent behaviors in iterative algorithms such as Fictitious Play [25]. The results in Figures 5–6 clearly show that also CFR can get stuck in non-convergent cycles, confirming what we observed in Figure 3. This is well known in theory (there is no guarantee of convergence for CFR in general-sum two-player games, even with no chance) but, to the best of out knowledge, was never observed in practice. Note that also CFR-S has some difficulties in reaching low values of $\varepsilon$, while CFR-Jr reaches a good approximation of an equilibrium point in few iterations.

Our extensive-form asymmetric variation of the Shapley game reads as follows:

- At each stage of the game, a player has to select a number in the set $\{0, 1, 2\}$.

- Player 1 selects a number and publicly discloses it. Then, player 2 chooses a number and writes it down, without disclosing it to the other player. Finally, player 1 selects another number, without knowing the previous choice of player 2.

- Let $s$ be the sum of the three numbers that have been selected. The players' utilities are computed as follows:

    - if $s \bmod 3 = 0$, then the utility is $(0, 0)$;

    - if $s \bmod 3 = 1$, then the utility is $(1, 0)$;

    - if $s \bmod 3 = 2$, then the utility is $(0, 1)$;

    - if the first number selected by player 1, and the number selected by player 2 are equal in value, then the utility gained by each player is doubled.

The last step is fundamental to introduce some asymmetries in the game and ensure that a uniform joint strategy (*i.e.*, $x(\sigma) = 1/|\Sigma|$, for each $\sigma$) is not a CCE. This is a problem in the standard Shapley game as many regret minimizers employ a uniform strategy as initialization, and therefore would converge *instantly* to a CCE.

Figure 5: Convergence in number of iterations for the Shapley game

Figure 6: Social welfare attained with respect to the optimal one for the Shapley game

### E.3 CFR-Jr with different joint distribution reconstruction rates

From a theoretical standpoint, CFR-Jr requires a joint distribution reconstruction step to be carried out at every iteration $t$, to ensure that the resulting normal-form joint strategy approaches the set of CCEs (see Theorem 5). We investigate whether it is possible to trade some accuracy for a reduction in computational time of the algorithm, by performing the joint distribution reconstruction at a subset of the iterations. This could also allow the algorithm to store smaller normal-form strategies, by skipping the reconstruction during the first iterations. Indeed, during the first iterations, CFR (and, therefore, CFR-Jr) returns behavioral strategies that tend to be fairly uniformly distributed over all the possible actions, leading to $\omega$ with a considerable number of strictly positive entries. This implies that the resulting normal-form strategies, in the first few iterations, have considerably large supports.

---

**Algorithm 4** CFR-Jr-$k$

---

1: **function** CFR-JR($\Gamma$)
2:     Initialize the joint strategy $\bar{x}$ to all zeros
3:     $t \leftarrow 0$
4:     **while** $t < T$ **do**
5:         **for all** $i \in \mathcal{P}$ **do**
6:             $\pi_i^t \leftarrow \text{CFR}(\Gamma, i)$
7:             **if** $t \bmod k = 0$ **then**
8:                 $x_i^t \leftarrow \text{NF-STRATEGY-RECONSTRUCTION}(\pi_i^t)$
9:         **if** $t \bmod k = 0$ **then**
10:             $\bar{x} \leftarrow \bar{x} + \bigotimes_{i \in \mathcal{P}} x_i^t$     $\triangleright \bigotimes_{i \in \mathcal{P}} x_i^t$ is joint distribution $x^t$ defined as the product of the players' normal-form strategies
11:         $t \leftarrow t + 1$
        **return** $\bar{x}^T = \frac{\bar{x}}{\left\lfloor \frac{T}{k} \right\rfloor}$

---

These considerations suggest that a slight modification of the CFR-Jr algorithm, that we call CFR-Jr-$k$ (see Algorithm 4), may perform better in some settings. The simple idea behind CFR-Jr-$k$ is that the reconstruction procedure is carried out only every $k$ iterations. We have evaluated CFR-Jr-$k$ for different values of $k$. In all the tests we performed, the CFR-Jr algorithm always showed good convergence. In Figures 7–9, we report the experimental results related to instances of Kuhn3-6. The plots show both the convergence speed in terms of number of iterations and in terms of run time, as well as the size of the support of the average joint strategy that was stored by the algorithm (which is always monotonically increasing by construction). In Figures 10–12, we report the experimental results related to instances of Kuhn3-10.

Larger reconstruction rates let the algorithm complete the same amount of iterations in a shorter time. On the other hand, smaller reconstruction rates can lead earlier to a good joint strategy, and hence to reach lower values of $\varepsilon$. There is a trade-off between iteration speed and reconstruction accuracy, which can be exploited to tackle different problems with the most suited level of precision.

For what regards the size of the support of the joint average strategy, we can clearly see that lower reconstruction rates, running more times the reconstruction algorithm in the same amount of time, and being more susceptible to high-frequency variations in the behavioral strategies built by CFR, require up to ten time more space to store their joint strategies.

Figure 7: K3-6. Convergence in number of iterations for CFR-Jr with different reconstruction rates

Figure 8: K3-6. Convergence in run time (seconds) for CFR-Jr with different reconstruction rates

Figure 9: K3-6. Size of the support of the joint strategy obtained from CFR-Jr with different reconstruction rates

Figure 10: K3-10. Convergence in number of iterations for CFR-Jr with different reconstruction rates

Figure 11: K3-10. Convergence in run time (seconds) for CFR-Jr with different reconstruction rates

Figure 12: K3-10. Size of the support of the joint strategy obtained from CFR-Jr with different reconstruction rates

## Footnotes

[9]Poly-APX is the class of optimization problems admitting a polynomial-time poly($\eta$)-approximation algorithm, where poly($\eta$) is a polynomial function of the input size $\eta$ [2].

[10] We denote by $\mathbf{0}$ a column vector of suitable dimension with all its elements equal to 0.