[Reviews · NeurIPS 2019]

Reviewer 1



The originality: It is interesting to learn the coarse correlated equilibrium of general-sum extensive form game with multi-player setting, and to reveal the computational complexity of the problem is important. From the technical point of view, some proof techniques are not so novel. For example, main parts of Theorem 3 follow the previous work [15]. The quality: Theorem 4 shows that the time complexity of Algoirhm2, which is a subroutine of CFR-Jr, and the time-complexity of CFR-Jr is not provided. In addition, there is no theorem that shows how good \epsilon CFR-Jr can provide. The clarity: It is well-written. The paper puts some appropriate examples and intuitive explanations. The significance: Theorems 2 and 3 seem to be important for machine learning and game theory communities. I cannot find how important Theorems 4 and 5 are. Below, I describe some questions mainly for the quality and the significance: (i) Does Theorem 3 mean that CFR-S does not necessarily converge to the optimal CCE? If so, it would be better to show the bound between the optimal CCE and a CCE to which CFR-S converges because an objective of the paper is to find a social-welfare-maximizing CCE. (ii) Theorem 4 provides a time complexity about the reconstruction. What is the time-complexity of CFR-Jr? (iii) At line 7 in Algorithm 2, if we put a pair (\bar{\sigma}_i,\bar{w}) and X already includes the same \bar{\sigma}_i, how do we build x_i from X? (iv) I cannot find if CFR-Jr approaches to the set of CCEs because Theorem 5 only indicates the relation between the regret and CCEs. There is no theorem showing how small \epsilon CFR-Jr can obtain. Thus, I want to know, given the number of iteration T, can we bound \epsilon parameterized by T? (v) In the experimental evaluation, how do you find the optimal social welfare SW_{Opt}?

Reviewer 2



I thoroughly enjoyed this paper. The paper contributes important results and algorithms with guarantees that are clearly the new state-of-the art. I have two relatively minor criticisms: - The algorithm CFR-Jr is rather straight-forward, as in, it is the obvious/natural way to get CFR to converge to approximate CCE, and the guarantees are a straight-forward consequence of properties of regret minimization. - CFR-S is the most unnatural CFR variant I have ever read. Why would one sample profiles if the values need to be updated in portions of the space that are not reachable anyway? One would think to resort to sampling because the problem is too large and to cut down the state space, but my understanding is that all the information states require updating even if they are not reached in the joint sample, which is why Theorem 3 can hold without any exploration. Is there a use case for this algorithm, when would it be preferred to CFR-Jr? There is oddly no discussion of EFCE. Does a similar notion not exist for CCE? [19] shows that dominated actions are removed by CFR. i would assume this is preserved in CFR-Jr; is that true? Or is it only be a property of the marginal strategies computed by CFR? If it's true, does it imply that the normal form CCEs filter out dominated strategies as well? It would be nice to see more graphs like the one in Figure 3 (across more of the domains), as it shows what the convergence and welfare looks like over time. If accepted for publication, I would encourage the authors to add more of these to the appendix.

Reviewer 3



**Originality** The complexity result is new, but the proof is quite similar to previously known results. The fact that the average of the strategies produced by CFR converges to CCE is well known and the goal of the algorithmic part of this paper is mainly to compactly represent the average of the strategy profiles from individual iterations. The transformation of behavioral to mixed strategy, which is the key component of the proposed method is interesting and I am not aware of a similar construction in previous literature. **Significance** Computing social welfare maximizing CCE is a relevant problem and proving its complexity is a reasonably significant result. I am unsure about the significance of the algorithmic results. The most natural version of the algorithm I could think of is to run CFR for T iterations and store the current behavioral strategies form all the iterations as the result. This should not require more than T|Z| storage. When playing, the mediator would just pick a random iteration and have the players play according to (a sample from) the behavior strategies from that iteration. The added value of CFR-Jr introduced in this paper seems to be only in transforming the result to a slightly more standard form. Furthermore, it is not obvious that the more standard representation of the strategy does not require more storage space than the naïve algorithm. The comparison of the sizes would better be shown theoretically, but it should be evaluated at least empirically. Similarly, I do not see any point in the CFR-S algorithm. It does not seem to be better than the naive solution or CFR-Jr in any aspect, so I am not sure what is the point in presenting it. It should be either clarified or the whole discussion of CFR-S should be removed. The experimental comparison of runtime is not surprising, since it compares a polynomial and exponential algorithm. However, the evaluation of the distance from the social welfare optimum is interesting. **Quality** The paper seems to be correct. My main concern is how the optimal CCE is computed in cases where CG did not finish and why are the results not reported for Kuhn poker even though all algorithms finished in time. The size of the final strategy representation should be reported in the experiments. **Clarity** The paper is reasonably clear and easy to read. However, there are several drawbacks: The paper is not very well motivated with respect to the naive algorithm described above. It is not very clear that the only concern for the algorithm is the compact representation of average strategy. It is not clear how the approximation of the social welfare is measured. **Questions** How is the optimal CCE computed in cases where CG runs for >24h. Is it still used? Why is the more exact time not reported? What is the typical size of the final solution in relation to the worst case, which seems to be T|Z|^2? What is the size of the normal formed mixed strategy profile compared to the original behavior strategy profile? ** After Rebuttal ** Thank you for your response. It helped to clarify my main concerns. I still believe that CFR-S is not worth the space in the paper and the paper should rather very carefully analyze the size of the computed solutions, at least to the extent of the rebuttal response. If this discussion is added, I would be for accepting the paper.

[Author Response · NeurIPS 2019]

[**Rev.1: on Th. 4 and 5**]. Th. 4 and 5 should be evaluated as the key building blocks of CFR-Jr, which is shown to outperform CFR and CFR-S in practice. Th. 4 is necessary to show the soundness of the reconstruction algorithm, which allows for working with compact normal-form strategies. Th. 5 shows that CFR-Jr approaches the set of CCEs, which is the goal of the algorithm. [**Rev.1:Q1**]. The reviewer is right. Indeed, almost all previous works on no-regret learning in games focus on the *easy* setting of two-player, zero-sum games, where optimality of the solution is always guaranteed. In multi-player, general-sum games, the problem of bounding the distance from the optimal equilibrium is a non-trivial open problem. As a first analysis, we provide some experimental evidence showing that we are able to approximate a nearly-socially-optimal equilibrium (e.g., Figures 3,5 and Tables 3,4). [**Rev.1:Q2**]. The time-complexity for one iteration of CFR-Jr is the same as the one for Alg. 2. CFR-Jr basically adds a traversal of the tree (requiring linear time), so the full time complexity is in the order of $T|Z|^2$. We will make this explicit in the paper. [**Rev.1:Q3**]. This can never happen, as plans $\bar{\sigma}$ built by the reconstruction procedure are always different (see the proof for Th. 4, Appendix D2, lines 632-635). [**Rev.1:Q4**]. The regret experienced by CFR-Jr is bounded by $\Delta|\mathcal{I}_i|\sqrt{|A_i|}/\sqrt{T}$, as in CFR [43], because our reconstruction procedure does not alter the way in which regret is minimized. We will make this explicit in the paper. [**Rev.1:Q5**]. We employed the optimal payoff (the maximum sum of players' utilities), which is not guaranteed to be achievable by a CCE, as an upper bound on the optimal social welfare. Employing an upper bound was necessary as CG could not scale to the larger instances of our test bed. We will clarify this in the paper. [**Rev.2:CFR-Jr**]. CFR-Jr has two fundamental components: a regret minimizer employing compactly representable strategies, and a poly-time reconstruction oracle (Alg. 2). The former comes almost directly from the available literature, but the latter was not obvious and is a key contribution of this paper. [**Rev.2: CFR-S**]. The sampling procedure has to take place at all information states because CFR-S needs a complete normal-form plan to keep track of the joint empirical frequency of play. Sampling paths of execution (from the root to a leaf) and recording them does not allow to build a well-defined correlation device, and the same holds for average marginal strategies (see Sec. 4.1 and Fig. 2). In general, CFR-S performs worse than CFR-Jr, as it needs much more iterations to converge. It has however a better per-iteration time complexity (linear in the input, as for vanilla CFR), and, thus, it could be preferred for very large game instances in which explicitly reconstructing the joint strategy (as in CFR-Jr) would be demanding. [**Rev.2: EFCEs**]. At the moment, there are no extensions of the CCE similar to the EFCE (we are aware of a team currently working on a notion of coarse EFCE). Providing a characterization of a new solution concept is beyond the scope of this paper. [**Rev.3: dominated actions**]. Properties shown in [19] should carry over to CFR-Jr as the core regret minimizer from CFR is left unchanged, and reconstruction of joint strategies does not introduce actions that were played with zero probability in the marginal strategies of the players. [**Rev.3: naïve algorithm**]. We explored the possibility of employing the reviewer's algorithm, but we decided not to include it for a number of reasons: i) We are interested in providing an explicit representation of the equilibrium (i.e., a joint probability distribution, as it is customary in the equilibrium computation community). In order to provide such a representation from $T$ behavioral strategy profiles, a reconstruction step would still be required, limiting the possible advantages of such approach. ii) In practice, CFR-Jr allows to build dramatically smaller solutions, e.g., the figure displays the percentage difference between CFR-Jr and the reviewer's algorithm storage spaces. The figure considers G2-4 with different tie-breaking rules.

iii) CFR-Jr is amenable to further improvements to reduce its storage space. E.g., we are currently working on heuristic approaches that perform the reconstruction step by prioritizing plans already selected in the past. These are part of the reasons for which we presented CFR-S as a naïve baseline (see next answer) instead of an algorithm analogous to the reviewer's one. [**Rev.3: CFR-S**]. Other than being used as a baseline, CFR-S is also instrumental in clarifying what needs to be added to vanilla CFR in order to find a CCE. Moreover, CFR-S is already faster than previous state-of-the-art approaches (based on MILPs). [**Rev.3: CG**]. See answer to Q.5 of Rev.1 for more details on how we compute the social welfare ratio. When we write >24h as the runtime of CG, we mean that the execution was killed by the time limit imposed on the system. The utility ratio is reported only for general-sum games

(in zero-sum games, e.g., Kuhn, the social welfare is always zero). [**Rev.3: size of the final solutions**]. In typical scenarios, the support of the joint strategies will often be the same among different iterations, so that the final solution size will be significantly smaller than the worst case. The figure shows the ratio against $T|Z|$, the ratio against the worst case would be even smaller as the denominator would be $T|Z|^2$. [**Rev.3: size of reconstructed strategies**]. The size of a reconstructed normal-form strategy profile is upper bounded by the number of leaves reached with strictly positive probability when following the original behavioral profile, which is, in its turn, upper bounded by the size of the support of the latter strategy (i.e., the number of non-zero action probabilities). Then, the reconstructed normal-form strategy for a single player has size upper bounded by the size of the original behavioral strategy. [**Rev.3: experimental evaluation**]. The reviewer claims that the experimental comparison of running times is not surprising. However, we remark that CFR-Jr has been compared with the current state-of-the-art technique for this setting (CG), and also with a new baseline algorithm (CFR-S). Moreover, CFR-Jr was also evaluated against the de facto standard in two-player games (CFR). CFR-Jr shows better convergence and attains a higher social welfare than all the other techniques.

[Meta-Review · NeurIPS 2019]

This paper proves that coarse correlated equilibria can be learned by a variant of CFR, called CFR-Jr. The main concerns of the reviewers are actually the CFR-S algorithm that seems quite naive and whose purpose is not clear. Is it just to make CFR-Jr looks better ? This could be a good "oral presentation" technique, but we believe that this paper would actually greatly benefit from being an important reorganization where CFR-S is more or less deleted (and replaced by interesting, short remarks, comments and discussion). We strongly suggest to the authors to perform this reorganization and incorporate the reviewers suggestions to the updated and revised version of the papers; it will definitely have a better impact that way.